# 🏠 AI Realtor: Towards Grounded Persuasive Language Generation for Automated Copywriting

## Abstract

This paper develops an agentic framework that employs large language models (LLMs) for grounded persuasive language generation in automated copywriting, with real estate marketing as a focal application. Our method is designed to align the generated content with user preferences while highlighting useful factual attributes. This agent consists of three key modules: (1) Grounding Module, mimicking expert human behavior to predict marketable features; (2) Personalization Module, aligning content with user preferences; (3) Marketing Module, ensuring factual accuracy and the inclusion of localized features. We conduct systematic human-subject experiments in the domain of real estate marketing, with a focus group of potential house buyers. The results demonstrate that marketing descriptions generated by our approach are preferred over those written by human experts by a clear margin while maintaining the same level of factual accuracy. Our findings suggest a promising agentic approach to automate large-scale targeted copywriting while ensuring factuality of content generation.

## 1 Introduction

While large language models (LLMs) have made significant strides across various tasks, their ability to persuade remains an underexplored frontier (see a discussion of related work in Section 6). This however is a particularly important capability since persuasion-related economic activities — a common thread in almost all voluntary transactions from advertising and lobbying to litigation and negotiation — underpin roughly 30% of the US GDP (Antioch, 2013), hence gives rise to tremendous opportunity for applying LLMs across a wide range of sectors. Meanwhile, this same potential introduces serious trustworthiness concerns. If LLMs can generate persuasive content at scale, their influence on human opinions raises risks of misinformation, manipulation and misuse, especially in sensitive domains such as political campaigns (Voelkel et al., 2023; Goldstein et al., 2024).

Therefore, we focus our study on the task of language generation for grounded persuasion, that is, the production of persuasive content that is faithful in factual details. This task is especially critical in copywriting, the practice of creating marketing text that seeks to influence consumer decisions, where its effectiveness can be directly assessed through measurable behavioral outcomes (e.g., ratings, engagement, and conversions), yet must remain strictly constrained by factual accuracy. In particular, we choose the domain of real estate marketing (see our rationale in § 2) and develop an agentic solution, 🏠 AI Realtor, under an economic scaffolding to investigate key elements of grounded persuasion. Below, we outline core contributions and the structure of this paper:

① **Real-World Evaluation**: Using real estate marketing as our testbed, we construct a large dataset from Zillow and design an experimental website that simulates the house search process, including buyer preference elicitation. We recruit a targeted group of potential home buyers to evaluate the persuasiveness of the generated marketing content (§ 2).

② **Theoretical Grounding**: We draw on the economic theory of information design in strategic communication games (Bergemann & Morris, 2019) to guide the agentic workflow. This includes processing the raw (factual) attributes of properties, selecting key features to highlight, and generating persuasive, human-like marketing content (§ 3).

③ **Agentic Pipeline**: We develop an LLM-based agent (§ 4) with three key modules: a *Grounding Module*, which mimics human expertise in identifying and signaling critical, credible selling points; a *Personalization Module*, which tailors content to user preferences; and a *Marketing Module*, which ensures factual consistency and incorporates localized features.

④ **Empirical Effectiveness**: Our system achieves a 70% win rate over human experts while maintaining, if not exceeding, the same level of factual accuracy, establishing the first LLM benchmark for grounded persuasion with measurable behavioral impact (§ 5).

## 2 A BENCHMARK FOR GROUNDED PERSUASION

**Motivations and Challenges**  Establishing a robust evaluation benchmark for persuasion faces two core challenges. First, persuasiveness is inherently subjective: unlike reasoning or planning (which have objective metrics), its effectiveness depends on human feedback and varies with individual preferences and contexts. Second, persuasion is multifaceted, with domain-specific techniques shaped by psychology, economics, and communication. Existing LLM research mostly focus on political or opinion-based persuasion, where evaluations are complicated by cognitive biases and adversarial framing. For example, Hackenburg & Margetts (2024) and Matz et al. (2024) reached conflicting conclusions using similar experimental designs. Durmus et al. (2024) highlight the anchoring effect – the tendency to cling to initial beliefs – making opinion shifts hard to measure. They also find fabricated content is often more persuasive, raising ethical and methodological concerns. These limitations underscore the need for new benchmarks in controlled, fact-grounded settings.

**Real Estate Marketing (REM) as Testbed**  Identifying well-scoped testbeds is key to launch systematic investigations of general AI capabilities, as demonstrated by recent benchmarks (Yao et al., 2022; Xie et al., 2024). The real estate marketing domain is ideal for our study because:

① *High-stakes, rational decisions*: Real estate involves high-stakes economic decisions, where buyers typically hold rational, fact-based beliefs — unlike more emotionally charged or polarized domains. Persuasive language in this setting must be both compelling and truthful.

② *Measurable economic impact*: Effective persuasion has tangible economic value in real estate. While structured attributes and images capture initial attention, industry guidance emphasizes that descriptive text is critical for conveying the unique experience of living in a home (Zillow, n.d.). The potential for LLMs to assist in this high-value task is further illustrated by recent anecdotal accounts (User, 2023).

③ *Rich, structured datasets*: The availability of extensive property listings with carefully labeled attributes (e.g., from Zillow) enables domain-specific training and thorough empirical evaluations.

**Realistic Evaluation Interface and Persuasiveness Measurement**  Our framework prioritizes two criteria: (1) immersive user interaction to capture authentic feedback and (2) dynamic preference elicitation for personalized generation. We replicate real-world homebuyer behavior by integrating 50k+ real-world listings into a web platform. See Appendix B and D for a full description of the web interface and dataset. We evaluate persuasion via pairwise comparisons: buyers view a property with two model-generated descriptions and select the more compelling one. Persuasiveness is quantified via Elo scores (Elo, 1967); factual accuracy is verified against listing metadata (see § 5).

## 3 AN ECONOMIC SCAFFOLDING OF COPYWRITING

Copywriting fundamentally is about communicating product information, often selectively, to shape potential buyers' perceptions and influence their purchasing decisions. This process of information signaling, also known as persuasion, has been extensively studied in decision theory and information economics (Spence, 1978; Arrow, 1996; Kamenica & Gentzkow, 2011; Connelly et al., 2011), typically within stylized mathematical models. To enable practical automated copywriting in natural language, we employ previous mathematical models/findings to build a framework compatible the agentic scaffolding enabled by modern language generation technology.

**Attributes**  Formally, we represent a generic *product* $X$ (e.g., a house or an Amazon item) as an $n$-dimensional vector $X = (X_1, X_2, \ldots, X_n)$. Each $X_i$ is called a raw attribute (or simply *attribute*). Attributes capture the factual and measurable characteristics of the product (e.g., square footage,

distance to transit). A specific product instance is denoted by vector $\mathbf{x} = (x_1, \cdots, x_n)$ where $x_i \in \mathcal{X}_i$ is the *realized* value of attribute $X_i$. Let $\mathcal{X} = \Pi_i \mathcal{X}_i$ be the domain of $\mathbf{x}$.

**Features**  Marketers often emphasize certain attractive properties of a product (e.g., "spacious layout" and "prime location" in REM), derived from its underlying raw attributes. We refer to these as signaling features (or simply *features*). Importantly, features differ from attributes: while some attributes may directly serve as features, features generally capture the more abstract (and sometimes ambiguous) properties. We denote the feature set as $S = (S_1, \cdots, S_m)$, with a feature vector $\mathbf{s} = (s_1, \cdots, s_m)$, where each $s_i \in [0, 1]$ quantifies the *intensity* or likelihood of feature $S_i$ being. For example, $S_i$ could be "bright room" and correspondingly $s_i$ denotes the extent to which rooms of the house are bright. In practice, both $x_i$ and $s_j$ can be assessed by domain experts.

**Signaling via the Attribute-Feature Mapping**  In our model, signaling features convey partial information to influence potential buyers' beliefs, leveraging the inherent cognitive mapping in natural language. For instance, a feature "bright room" may probabilistically imply high floor, southern exposure, and modern lighting – all affecting buyers' perceptions and decisions. (e.g., deciding to schedule a visit). We formalize this with a mapping $\pi : \mathcal{X} \to [0, 1]^m$ that transform raw attributes $\mathbf{x} \in \mathcal{X}$ into feature intensities $\mathbf{s} \in [0, 1]^m$. That is, $\mathbf{s} = \pi(\mathbf{x})$. Sometime, we use $\mathbf{s}(\mathbf{x})$ to emphasize the dependence of $\mathbf{s}$ on the underlying attributes $\mathbf{x}$, and $s_j(\mathbf{x})$ is its $j$-th entry. This mapping reflects the commonsense inference: given $\mathbf{x}$, how strongly we can claim the presence of feature $S_j$.

This attribute-feature mapping $\pi$ is widely studied in both machine learning and economics. In Bayesian statistics, $X_i$ is an observable variable, $S_j$ a latent variable, and $\pi$ captures their probabilistic dependence. In information economics, $X_i$ represents a state, $S_j$ a *signal*, and $\pi$ is known as a *signaling scheme*. Signals can be strategically designed to reveal partial information about the state, and prior work has made significant progress in their optimal design to influence the equilibrium outcomes (Kamenica & Gentzkow, 2011; Bergemann et al., 2015; Bergemann & Morris, 2019). Our work moves beyond this traditional Bayesian framing to incorporate the nuanced role of natural language–often abstracted away in prior models–and to uncover the implicit, *commonsense* mappings behind linguistic signals, rather than design new schemes.

**Marketing Design under Information Asymmetry**  Marketing fundamentally exploits information asymmetry between sellers and buyers (Grossman, 1981; Lewis, 2011; Dimoka et al., 2012; Kurlat & Scheuer, 2021). This important insight, along with its broader implications in general economic markets, was notably recognized by the 2002 Nobel Economics Prize (Akerlof, 1978; Spence, 1978; Stiglitz, 1975; Löfgren et al., 2002). In our setting, the seller or seller's agent knows the exact product attributes $\mathbf{x}$ and the corresponding feature values $\mathbf{s}(\mathbf{x})$, while the buyer enters the market with only a prior belief $\mu$ over the distribution of attributes in $\mathcal{X}$. Without specific knowledge of the product $\mathbf{x}$, the buyer holds an expected belief over features:

$$\text{Initial belief of features:} \quad \bar{\mathbf{s}}(\mu) = \int_{\mathbf{x} \in \mathcal{X}} \mathbf{s}(\mathbf{x}) d\mu(\mathbf{x}). \tag{1}$$

Given the asymmetric feature beliefs between the buyer and seller, the purpose of marketing can be described as revealing features, subject to communication constraints, to shift the buyer's belief from $\bar{\mathbf{s}}(\mu)$ towards $\mathbf{s}(\mathbf{x})$ with the goal of increasing the product's attractiveness to the buyer.

**Grounded Persuasion in Natural Language**  The remaining part of our model is to optimize the persuasiveness of marketing content. The typical approach in economic theory is to develop models capturing buyers' belief updates and decision-making processes. However, these are difficult to operationalize due to the absence of concrete buyer utility functions and behavioral models. Instead, we leverage the generative capabilities of LLMs, guided by heuristics and instructions tailored for grounded persuasion. At a high level, we use the attribute-feature mapping $\pi$ to guide the selection of a feature subset $\mathcal{S}^*$ to emphasize in generation. User preferences $\mathbf{r}$ are elicited and incorporated into a prompt $\mathcal{I}^*$ for personalization. We hypothesize that the LLM approximates the solution to an implicit optimization problem: $L^* = \arg\max_{L \in \mathcal{L}} \Pr(L | \mathcal{I}^*, \mathcal{S}^*, \mathbf{r}) \approx \arg\max_{L \in \mathcal{L}(\mathbf{x})} U^{\mathbf{r}}(L)$. That is, the language $L^*$, output by an LLM provided carefully designed prompts $\mathcal{I}^*$, selected features $\mathcal{S}^*$ and user preferences $\mathbf{r}$, could approximately maximize users' preference-adjusted persuasiveness function $U^{\mathbf{r}}$. Moreover, the generated language $L$ will obey product facts (i.e., is *grounded*), or concretely, be drawn from set $\mathcal{L}(\mathbf{x})$ that includes all languages consistent with the product attribute $\mathbf{x}$. Our subsequent agent implementation and its practical effectiveness support this hypothesis; we further conjecture that more powerful models will generally be able to find better-approximated

solutions to this optimization problem. Given this formulation, our design objective is to support the LLM in solving the above optimization problem by constructing effective prompts $\mathcal{I}^*$, selecting appropriate features $\mathcal{S}^*$, and representing user preferences $\mathbf{r}$. The following section describes our implementation.

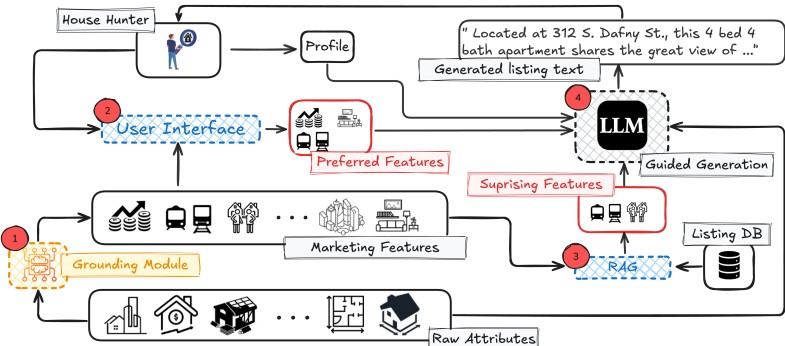

Figure 1: Illustration of the Design Pipeline of 🐱 AI Realtor.

# 4 THE AGENTIC IMPLEMENTATION OF 🐱 AI REALTOR

This section outlines the core design of 🐱 AI Realtor, an AI agent that process multiple levels of marketing information to compose persuasive descriptions for real estate listings and actively learn to adapt its language to individual buyer preferences. At a high level, our approach operationalizes microeconomic models by implementing the following three key ingredients:

- Grounding Module: identify the attribute-feature mapping $\pi$;
- Personalization Module: elicit and represent buyer preferences $\mathbf{r}$;
- Marketing Module: select useful yet factual marketing features $\mathcal{S}^*$ based on $\pi, \mathbf{r}$.

The overall system pipeline is illustrated in Figure 1. Below, we highlight the novel contributions within each of the three modules. Full implementation details are provided in Appendix C.

## 4.1 GROUNDING MODULE: PREDICTING CREDIBLE FEATURES FOR MARKETING

Our model assumes the existence of attribute-feature mappings that marketers can use to influence buyer beliefs and behaviors. However, a key challenge is that while raw attributes (e.g., square footage) are available, high-level signaling features (e.g., "convenient transportation") lack explicit annotations in our dataset. This absence of supervision, combined with the open-ended nature of natural language, where many tokens may serve as features with overlapping or ambiguous meanings, makes the learning problem inherently difficult. Without a structured representation, the label space becomes too sparse for effective training. Indeed, we find that directly prompting LLMs to generate features produces redundant or incomplete feature sets, which undermines the quality of the learned mapping.

Manual annotation by human experts could address this issue but is labor-intensive, costly to scale, and difficult to personalize. We therefore adopt a machine learning approach to infer the attribute-feature mapping automatically from unlabeled data, guided by LLM-assisted schema construction and weak supervision. Specifically, we provide LLMs with a large pool of candidate features extracted from the dataset and prompt them to organize these into a hierarchical schema. A small number of human annotators validate the output to monitor hallucinations and refine definitions. This process, illustrated in Figure 2, yields a compact and expressive feature representation. Once created, this feature set and mapping can be reused across models within the same marketing domain and is thus a *one-time* cost.

Using the finalized feature schema, we guide an LLM to annotate whether each feature $s_i$ is present in a given listing, based on its attributes $\mathbf{x}$ and corresponding human-written description. After standard preprocessing (e.g., removing low-quality texts, normalizing attributes), we curate a labeled

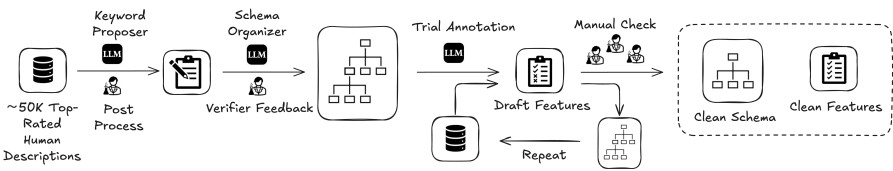

Figure 2: Illustration of the inductive feature schema construction pipeline.

dataset and train a neural network to learn the attribute-feature mapping.[1] On a random 4:1 train-test split, our model achieves 69.39% accuracy and 67.43% F1 score. This accuracy is already high, given the large amount of available features and stochastic nature of the signaling process.

To ensure grounded use of signaling features, we implement a deterministic feature selection strategy: only features with intensity $s_j \geq \alpha$ are retained. In our implementation, we use the threshold $\alpha = 1/2$[2] and define the resulting set of *marketable features* as:

$$\text{Marketable Features:} \quad \mathcal{S}_1(\mathbf{x}) = \{S_j : s_j(\mathbf{x}) \geq \alpha\}. \tag{2}$$

### 4.2 PERSONALIZATION MODULE: ALIGNING WITH PREFERENCES

This stage aims to steer persuasive language generation toward buyer preferences—another core objective of grounded persuasion. Our solution involves two steps.

First, we elicit user preferences and structure them in a usable form. On platforms like Zillow or Redfin, this could be done using mature machine learning methods based on user browsing behavior. Without access to such data, we instead design a preference elicitation process within our human-subject evaluation framework. Specifically, our web interface prompts an LLM to simulate a realtor, guiding participants through questions to identify their most valued features. Each user then rates the importance of each feature $S_j$ with a score $r_j$ prior to the evaluation tasks. While simple, this approach suffices to support a persuasive AI Realtor that effectively adapts to user preferences, as demonstrated in our experiments.

Second, we select a personalized subset of features to shift user beliefs positively. Since real-world marketing texts are not tailored to individual users, we cannot rely on them to provide supervision for personalization. Instead, we use a scoring function that combines population-level feature intensity $\mathbf{s}(\mathbf{x})$ with individual preference ratings $\mathbf{r}$, selecting features above a threshold $\alpha$:

$$\text{Personalized Features:} \quad \mathcal{S}_2(\mathbf{x}) = \{s_j \mid s_j(\mathbf{x}) + c(r_j - r_0) \geq \alpha\},$$

where $c$ reflects the strength of personalization and $r_0$ is a baseline rating. These features are then passed to the LLM, which determines how best to incorporate them into the generated text.

### 4.3 MARKETING MODULE: CAPTURING SURPRISAL VIA RAG

The last stage is designed to better ground persuasive language generation in factual evidence, problem contexts and localized information in automated marketing. Our design here is inspired by rich marketing strategy research (Lindgreen & Vanhamme, 2005; Ludden et al., 2008; Ely et al., 2015), which have shown that buyers would derive entertainment utility from *surprising* effects/features and have a deeper impression. In our setting of real estate marketing, such surprising features are those that are relatively rare compared to their surrounding area. Formally, we determine a set of surprising features based on their percentile in the feature distribution as follows,

$$\text{Surprising Features:} \quad \mathcal{S}_3(\mathbf{x}) = \{S_j \subset \mathcal{S}_1 : s_j(\mathbf{x}) \text{ is within } \beta\text{-quantile of distribution } s_j(\mu)\}.$$

---

[1] We also experiment with several other baselines for feature extraction, including prompting LLMs directly and applying simple pooling over embedding vectors. The strongest baseline achieves approximately 59% F1 score, which is substantially lower than the final model used in our grounding module. For simplicity, we only reported the final model's performance in the main text.

[2] The feature existence threshold $\alpha$ was determined through a grid search over the range $[0.1, \ldots, 0.9]$, with performance evaluated using the F1 score on a held-out, human-annotated validation set. $\alpha = 0.5$ yielded the best trade-off between precision and recall.

This gives the LLMs localized feature information at different levels of granularity obtained through Retrieval Augmented Generation (RAG) (Lewis et al., 2020).[3] Such behavioral economics-driven design proves to be highly effective; citing one of the human subjects in our experiment (see the full description in Appendix A.1), who was asked about why they liked a listing description (without knowing it was AI-generated):

> ...*Description B specifically points out the rarity of the ample storage and built-in cabinetry in similarly priced listings, making the property stand out.*

## 5 EVALUATIONS

### 5.1 EVALUATION BY HUMAN FEEDBACK

To evaluate the effectiveness of listing descriptions generated by different models, we draw inspiration from ChatArena (Zheng et al., 2023) and conduct an online survey to collect pairwise human feedback comparing different models' outputs. In summary, systematic evaluation by human feedback shows that our 🏠 AI Realtor clearly outperforms human experts and other model variants, measured by standard Elo ratings (Elo, 1967). Below, we detail the design of our user survey platform, baseline setup, and evaluation metrics, followed by a report on the human evaluation results.

**Quality Assurance**  We focus on the major US city *Chicago*[4] with a highly active housing market. We recruit about 100 participants from the popular *Prolific* platform for human-subject experiments, selecting in-state residents familiar with Chicago's housing market and curating approximately 1,000 listings of varied sizes and price ranges. Each human subject is tasked with comparing 10 pairs of house descriptions. During each comparison, the human subject sees pictures and all basic information about a house, and then faces two listing descriptions without knowing what methods (human realtor or AI agents) generate them, and is asked to choose which description is preferred, and by how much (see Appendix B.3 for details). Notably, 🏠 AI Realtor generates personalized descriptions on the fly for each human subject, based on their preferences elicited while they join the survey (see Appendix B.2 for details).

To ensure feedback quality, we implement several measures: (1) *Screening tests* to confirm participants can extract information from listings and follow specific home search motives (See Appendix B.1 for details); (2) *Attention checks* using pairs of nearly identical descriptions to ensure participants carefully compare and identify differences; (3) *Control experiments* where participants compare human-written, engaging descriptions against LLM-generated descriptions intentionally prompted to be plain and unappealing, verifying their ability to favor high-quality descriptions; and (4) *Incentives* on the platform, including bonus payments and requests for written reasoning behind choices, to encourage consistent, well-justified feedback.

**Metrics**  We adopt the Elo rating score as our main metric. We use a typical choice of the initial Elo rating as 1000, scaling parameter $c = 400$, and learning rate $K = 32$. The win rate for a model with Elo rating $e_1$ against a model with rating $e_0$ is calculated as $\left[1 + 10^{(e_0 - e_1)/c}\right]^{-1}$.

**Baseline Models**  In addition to our primary persuasion model 🏠 AI Realtor, we evaluate several baseline models, including: *Vanilla*, an LLM prompted with all attributes of the listing; *SFT*, an LLM fine-tuned with supervised training and prompted with all features of the listing; *Human*, listing descriptions sourced from Zillow, written by professional realtors; *Control*, the model used in the control experiment described earlier. We also include two ablation models based on 🏠 AI Realtor: one that only uses the marketable feature from the Grounding module, the other excludes surprisal features from the Marketing module. Additionally, we experiment with two LLM variants, GPT-4o and GPT-4o-mini, while keeping the prompt instructions consistent across models.

---

[3]In our implementation, we implement the sparse retrieval part via ElasticSearch (https://www.elastic.co/elasticsearch) and retrieve Top 10 listings with the most similar features.

[4]Chicago has been established by various economic and sociological literature (Levitt & Syverson, 2008; Sampson, 2012; Grabinsky & Reeves, 2015) as a rigorous proxy for broader American urban mechanics. Also, Chicago has a diverse set of listings, compared to major cities in the US, that can reliably test our models' performance across various scenarios. See Appendix D.3 for more analysis.

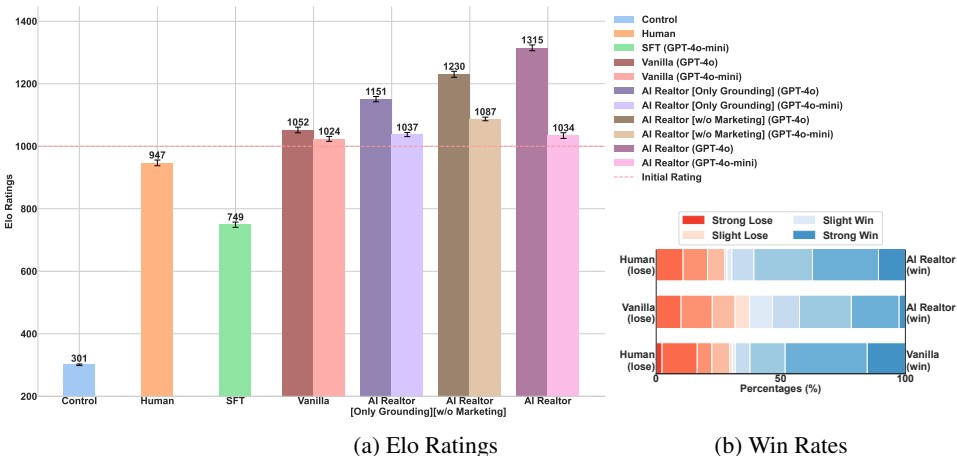

(a) Elo Ratings  (b) Win Rates

Figure 3: Comparison of model performance using Elo ratings and win rates. Elo ratings represent overall persuasiveness, and win rates reflect relative persuasiveness. Both metrics are based on evaluations by human subjects. Confidence Interval is computed based on 500 bootstrap runs by adapting the Elo implementation from Chatbot Arena (Chiang et al., 2024)

**Results** We plot the Elo ratings of different models in Figure 3a. The results reflect a clear trend: while vanilla GPT-4o performs on par with humans (1052 vs 947), each of our designed module enhancement progressively improves the persuasiveness of the generation, ultimately surpassing human performance with a clear margin (1318 vs 947). To ensure a fair comparison against human descriptions, which do not have access to explicit user preferences, we note that our model variant without any personalization (*Only Grounding*) still significantly outperforms human-written content (1151 vs 947). Also we observe that using GPT-4o to generate listing description does have a clear edge compared to that of GPT-4o-mini. Moreover, we plot empirical win rates among three major competitors (*Vanilla*, *Human* and 🏠 AI Realtor) in Figure 3b, which directly illustrates how much 🏠 AI Realtor outperforms the other two.[5] Please see Appendix A for case studies of our model-generated descriptions with more nuanced observations.

## 5.2 EVALUATION THROUGH AI FEEDBACK

Human feedback can be costly, especially as we scale the training and evaluation of our task. In this section, we report our empirical evaluation by using AIs to simulate human feedback based on our data collected from the above human-subject experiments.

**Simulation Setup** We employ an LLM to simulate the responses of buyers in the previous experiment. We use the first $K$ pairwise comparison results as $K$-shot in-context learning samples and prompt the LLM to predict the same buyer's selections for the remaining samples. We also adopt the chain-of-thought prompting format (Wei et al., 2022) and provide the buyer's rationale comments as the information for in-context learning (see Appendix F.7 for the exact prompt). We use the Sotopia framework (Zhou et al., 2024) to configure this simulation agent with GPT-4o-mini (OpenAI, 2024b) as the base model.

**Metrics** We use two metrics to evaluate the reliability of AI feedback compared to human feedback: 1) *Shot-wise Simulation Accuracy (SSA)*: the prediction accuracy averaged across users for each shot; 2) *User-wise Simulation Accuracy (USA)*: the prediction accuracy for each user, averaged across #(shots). The first metric measures overall simulation accuracy across the entire population, while the second one measures simulation accuracy for each user.

**Effectiveness of AI Feedback** The simulation results under both metrics are shown in Figure 4a and 4b. The model achieves 61.6% accuracy across users and exhibits non-trivial ($> 50\%$) performance for 79.2% of users, suggesting potential for leveraging AI feedback. However, the accuracy remains unsatisfactory for reliable evaluation. Additionally, the variance in the USA metric is high and increases with more provided shots, underscoring the challenges of personality simulation, as

---

[5]Participants also rated their preference for each description on a 1-5 scale.

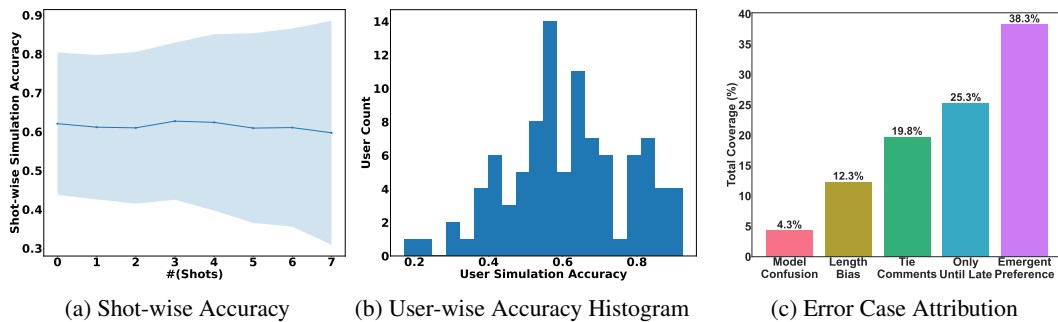

| (a) Shot-wise Accuracy | (b) User-wise Accuracy Histogram | (c) Error Case Attribution |

Figure 4: Analyses of Simulating Human Feedback with AI Feedback.

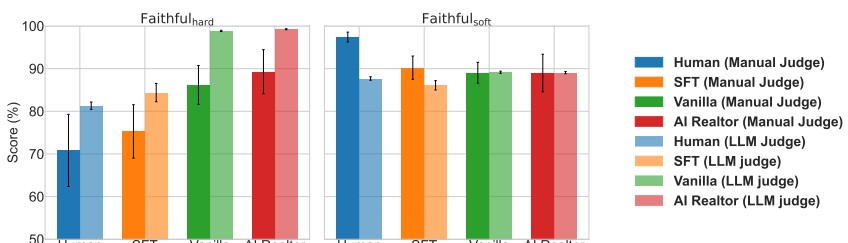

Figure 5: Faithfulness Scores for Hallucination Checks.

highlighted in (Wang et al., 2024). While the upward trend in variance is expected due to fewer data points, it highlights the difficulty of predicting user preferences dynamically.

To further understand the limitations of AI-simulated feedback, we conduct a manual analysis of simulation errors. Excluding the $56.1\%$ error cases that lack clearly explainable patterns, we attribute the rest of them to several key error sources in Figure 4c: 1) *Length Bias*: Similar to the observation in Chatbot Arena (Zheng et al., 2023), the model overly favors longer responses; 2) *Tie Comments*: Buyers consider the influence from descriptions as indifferent yet still cast confident votes in one of the choices; 3) *Emergent Preference*: While the model only has access to a buyer's pre-established preference, a buyer's selections in some cases reflect some unspecified preferences or ones in contradiction; 4) *Only Until Late*: Correct predictions about a buyer's selection only emerge after sufficient in-context samples; 5) *Model Confusion*: The model's prediction appears random, which indicates that the model may not have sufficient information to simulate such a buyer. Some of these errors can be mitigated by collecting more selection data from each buyer or improving the preference elicitation process in future work.

## 5.3 HALLUCINATION CHECKS

For grounded persuasion, it is important to ensure minimal risks of hallucination. Hence, we evaluate the amount of misinformation in the marketing content through fine-grained fact-checking (Min et al., 2023), where we use GPT-4o to assist our hallucination check and set the listing attributes in the dataset as atomic facts. Specifically, we consider two types of factual attributes to check, $X_{\text{hard}}$ and $X_{\text{soft}}$. For attributes in $X_{\text{hard}}$, we require the attribute description to be completely accurate (e.g., #(bathrooms)), whereas we allow attributes in $X_{\text{soft}}$ to be roughly accurate (e.g., address).

Given an attribute set $X$ and a description $L$, we ask the model to perform the following tasks: $\text{supp}(L, X)$ identifies the subset of attributes in $X$ that are mentioned in $L$; $\text{eval}_{\text{hard}}(L, x)$ returns a binary value indicating whether attribute $x$ is accurately described; and $\text{eval}_{\text{soft}}(L, x)$ provides a score from 0 to 10 reflecting the extent to which $x$ is accurately described (see our prompt design in Appendix E). We then compute the faithfulness score for attributes in $X_{\text{hard}}$ and $X_{\text{soft}}$ as follows,

$$\text{Faithful}_{\text{hard}}(L) = \frac{\sum_{x \in \text{supp}(L, X_{\text{hard}})} \text{eval}_{\text{hard}}(L, x)}{|\text{supp}(L, X_{\text{hard}})|}, \quad \text{Faithful}_{\text{soft}}(L) = \frac{\sum_{x \in \text{supp}(L, X_{\text{soft}})} \text{eval}_{\text{soft}}(L, x)/10}{|\text{supp}(L, X_{\text{soft}})|}.$$

As shown in Figure 5, the model-generated descriptions are mostly faithful to listing information with minimal hallucination under both metrics. In contrast, the descriptions from human realtors or SFT model show an even higher level of hallucination. After digging into details, we found that this is due to human realtors' (also SFT's) vague description of attributes in $X_{\text{hard}}$ such as the following example, "*This 4 bedroom, 3.5 bathroom home offers nearly 2,000 (1,828) sqft of living space...*". Our 🏠 AI Realtor, however, tends to accurately describe factual attributes whenever mentioned, likely due to its preference to copy from context — interestingly, this preference seems to be forgotten by the model after supervised fine-tuning on human-written descriptions. That said, it is debatable whether such vague descriptions of attributes is a true kind of hallucination, though some buyers did complain about this kind of language in the comments of their responses.

We replicate hallucination checks with human evaluators to validate GPT-4o's hallucination detection results. Details of the interface and annotation guidelines are provided in Appendix E.2, and the results are shown in Figure 5. Regarding ranking consistency, GPT-4o's relative ordering of models on $X_{\text{hard}}$ aligns closely with human evaluations, but diverges on $X_{\text{soft}}$, highlighting the challenge of verifying loosely matched factual attributes. Overall, both human and GPT-4o evaluations show that 🏠 AI Realtor achieves higher faithfulness on $X_{\text{hard}}$ and comparable performance on $X_{\text{soft}}$, suggesting it poses minimal risk of hallucination. Furthermore, the human evaluators report that 🏠 AI Realtor descriptions are as trustworthy as humans (See more details of our credibility survey in Appendix E.2).

## 6 RELATED WORK

Several studies have pioneered methods in computational linguistics for understanding and measuring persuasiveness (Wang et al., 2019; Wei et al., 2016; Tan et al., 2016). The advent of large language models (LLMs) has further spurred research into their persuasive capabilities, especially as part of frontier model risk assessments by developers (Durmus et al., 2024; Hurst et al., 2024; Jaech et al., 2024). A major focus has been on the potential for LLM-generated propaganda in politically sensitive contexts (Voelkel et al., 2023; Goldstein et al., 2024; Hackenburg et al., 2024; Luciano, 2024). Parallel investigations examine settings such as personalized persuasion (Hackenburg & Margetts, 2024; Salvi et al., 2024; Matz et al., 2024). Breum et al. (2024) and multi-round persuasion (Breum et al., 2024). Takayanagi et al. (2025) assess the influence of GPT-4's ability to generate financial analyses to audiences. Complementary research has probed related LLM capabilities including negotiation (Bianchi et al., 2024), debate (Khan et al., 2024), sycophancy (Sharma et al., 2023; Denison et al., 2024), as well as the emergence of strategic rationality in game-theoretic settings (Chen et al., 2023; Raman et al., 2024).

In a similar application domain, Angelopoulos et al. (2024) conduct an experiment to generate marketing email with a fine-tuned LLM and report a 33% improvement in email click-through rates compared to human expert baselines. Singh et al. (2024) design an evaluation benchmark based on a dataset of tweet pairs with similar content but different wording and like counts. In comparison, our work develops a full agentic solution for automated marketing from learning domain expert knowledge to crafting localized features, which significantly outperforms the model with supervised fine-tuning in our human-subject experiments.

## 7 DISCUSSION

**Contributions and Implications** This paper presents a novel framework for persuasive language generation, marking a first step toward integrating signaling schemes from economic theory into agentic LLM design. Our results demonstrate that this structured approach can achieve superhuman persuasive performance in a high-stakes domain like real estate marketing. A central tenet of our design is the deliberate prioritization of factual grounding. While human-written descriptions often employ stylized or emotionally resonant language, we argue that in domains where accuracy is paramount, constraining generation to verifiable facts is a necessary and responsible choice. Our framework's effectiveness stems from its ability to map raw attributes to a compact set of high-level, market-relevant features, ensuring that the generated content is both persuasive and credible.

**Limitations and Future Directions** Despite these promising results, we acknowledge several limitations that highlight avenues for future research. The primary bottleneck remains the reliance on

high-quality human feedback for evaluation. Our experiments with automated, LLM-based evaluators show promise for assessing factuality but are not yet reliable for measuring nuanced qualities like persuasiveness, underscoring the need for more sophisticated evaluation benchmarks. Second, our empirical validation is currently concentrated on the Chicago market. While our agentic workflow is designed to admit localization, persuasion is culturally and economically sensitive; future work is required to verify stability across diverse geographic regions and demographics. Furthermore, generalizing this framework to domains with less structured inputs or different persuasive norms (e.g., brand marketing vs. legal arguments) presents a significant and important challenge.

Building on this foundation, several exciting directions emerge. The modularity of our framework is well-suited for incorporating domain-specific constraints. For regulated fields like housing or finance, integrating compliance filters or legal principles inspired by approaches like Constitutional AI (Bai et al., 2022) is a crucial next step to ensure responsible deployment. Moreover, to address the trade-off between factuality and expressiveness, future work could explore incorporating a wider range of persuasion theories, such as emotional appeals and narrative structures, as controllable modules within the agentic design. Finally, scaling our datasets, expanding to new copywriting domains, and conducting more extensive real-world A/B testing will be essential to fully unlock the potential of theory-grounded persuasive generation.

## ETHICS STATEMENT

Our research on persuasive language generation acknowledges the dual-use nature of such technologies. We have proactively centered our work on grounded persuasion, where generated content is constrained by verifiable facts, to mitigate the risks of misinformation and manipulation. Our extensive hallucination checks, detailed in § 5.3 and Appendix E, confirm that our agent maintains a high degree of factual accuracy, comparable to or exceeding that of human experts.

All human-subject experiments were conducted in compliance with ethical research standards. The study protocol received IRB approval (exempt). Participants were recruited from the Prolific platform, informed of the study's purpose, and compensated at a fair rate (approximately $20/hour with performance incentives). The dataset, derived from publicly available Zillow listings, was processed to remove any personally identifiable information, ensuring user privacy.

By focusing on a high-stakes, fact-driven domain like real estate, we aim to provide a framework for developing responsible persuasive AI. We believe this work serves as a foundation for future research into the ethical guardrails necessary for deploying strategic language models in real-world applications and encourage continued investigation into their broader societal implications.

## REPRODUCIBILITY STATEMENT

We are committed to ensuring the reproducibility of our research. Below, we outline the resources available to replicate our findings.

**Data.** The core dataset was constructed from publicly available real estate listings from Zillow. The raw attribute schema, data curation process, and final feature schema are detailed in Appendix D. The collected human-subject evaluation data and feature annotations will be made publicly available upon publication.

**Methodology and Code.** The theoretical framework is described in § 3. The complete agentic pipeline, including the implementation of the Grounding, Personalization, and Marketing modules, is detailed in § 4 and Appendix C. To facilitate replication, all prompts used for LLM-based feature extraction, schema generation, and persuasive content creation are provided in Appendix F. The full source code for our agent and evaluation framework will be released publicly.

**Evaluation.** Our human-subject evaluation framework, including the design of the web interface, participant screening, and preference elicitation process, is fully described in § 5.1 and Appendix B. The metrics used, including Elo rating calculations and hallucination checks, are also detailed in § 5.

## USAGE OF LARGE LANGUAGE MODELS

In this work, we mainly use LLMs for the following purposes:

1. Aid or Polish Writing (Gemini 2.5 Pro, ChatGPT 4/5)
2. Literature Retrieval and Discovery (e.g., finding related work) (Gemini 2.5 Pro Deep Research, ChatGPT Deep Research)
3. Assisting Code Writing and Debugging (Claude 3.5 Sonnet)

We fully understand the responsibility of using LLMs in academic research. We carefully monitor any potential problems, such as plagiarism or scientific misconduct (e.g., fabrication of facts) when using LLMs. We make sure these problems do not occur in the paper.

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

## A  CASE STUDIES

### A.1  USER FEEDBACK ON GENERATED DESCRIPTIONS WITH SURPRISAL FEATURES

Below is an example where the user considers the description generated by 🐻 AI Realtor to be more persuasive, specifically because it includes surprisal features.

> **Disfavored Description (🐻 AI Realtor [w/o Grounding], Description A):** Welcome to your dream condo in the vibrant Printers Row neighborhood! This stunning one-bedroom, one-bathroom home is not only affordable but packed with modern amenities and stylish features, all

for $297,500. Nestled in the heart of Chicago, 547 S Clark St APT 405 offers unparalleled convenience with easy access to public transportation, shopping, dining, and the iconic Millennium Park. Built in 1998, this condo boasts a huge bedroom suite, hardwood flooring throughout, and an inviting gas fireplace. The newly upgraded stainless steel appliances and eye-catching granite countertops make the kitchen a chef's delight. Enjoy your morning coffee on the spacious private balcony, while additional storage and a walk-thru closet area cater to all your organizational needs. The property is equipped with central air and in-unit laundry for ultimate comfort and convenience. With a $389 monthly HOA fee, this condo promises both affordability and a secure investment in the bustling heart of Chicago. Don't miss the opportunity to embrace a lifestyle of ease and style in this perfect one-bedroom sanctuary!

**Preferred Description (🧑‍💼 AI Realtor, Description B):** Discover the splendid charm of urban living at 547 S Clark St APT 405, a chic one-bedroom, one-bathroom condo in the heart of Chicago's desirable Printers Row. Priced at an enticing $297,500, this condo harmonizes affordability with luxury, featuring an expansive 876 sqft layout adorned with elegant hardwood floors throughout and a cozy gas fireplace. Enjoy culinary delights with new stainless steel appliances and eye-catching granite countertops in the kitchen. The huge bedroom suite boasts a walk-thru closet area, offering ample built-in cabinet space and additional storage – an rarity in similarly priced listings. Revel in the tranquility of your spacious private balcony, perfect for unwinding with views of the bustling cityscape. Keep convenience at your fingertips with in-unit side-by-side laundry and parking in a secure garage. Built in 1998, this condo is a stone's throw away from Millennium Park and surrounded by public transportation, grocery stores, and cultural amenities, amplifying its urban attraction. With a manageable monthly HOA fee of $389 and strong investment potential, this condo ensures financial peace of mind alongside lifestyle enrichment. Don't miss out on this prime opportunity in vibrant Cook County, where your ideal location meets impeccable structure and style.

**User Comment:** I selected Description B because while both descriptions effectively showcase the property's appealing features, Description B employs a more evocative and sensory language, creating a stronger emotional connection and a more vivid image of the lifestyle the condo offers. Specific points of comparison:
- Language & Tone:
Description B uses words like "splendid charm," "chic," and "harmonizes" to paint a picture of elegance and sophistication, creating a more aspirational tone.
Description A, while positive, uses more straightforward language, focusing on practicality and convenience.
- Emphasis on Space & Luxury:
Description B highlights the "expansive 876 sqft layout," and the "huge bedroom suite," emphasizing the sense of space and luxury.
Description A also mentions the spaciousness but doesn't create as strong an image of grandeur
- Unique Selling Points:
Description B specifically points out the rarity of the ample storage and built-in cabinetry in similarly priced listings, making the property stand out.
Description A focuses on the general convenience and modern amenities, which, while attractive, are not as unique.
- Lifestyle & Surroundings:
Description B paints a more vivid picture of the lifestyle the condo offers, inviting the buyer to "revel in the tranquility" of the balcony and highlighting the proximity to cultural amenities, creating a stronger sense of place.
Description A mentions the location and amenities but lacks the same level of detail and emotional connection.
- Overall:
Both descriptions are well-written and informative, but Description B's richer language, focus on unique features, and emphasis on lifestyle create a more compelling and emotionally resonant picture of the property. It makes the condo feel more desirable and aspirational, which is likely to attract more potential buyers.

Meanwhile, we acknowledge that it is also challenging to generate appropriate language to express surprising features. Here is an example that confuses the user:

**Preferred Description (Human, Description A)**: Step into a light-filled sanctuary where high ceilings and expansive windows create an open and airy atmosphere. The loft-style design seamlessly combines living and dining spaces, providing a versatile canvas for your personal style. Ascend to your personal rooftop oasis, where the cityscape becomes your backdrop. Whether you're hosting soirees under the stars or enjoying a quiet morning coffee, this terrace is a true escape from the ordinary. Features include hardwood floors throughout, in-unit washer and dryer and new heat/central air conditioning. Secure building with intercom system, a deeded garage parking space and large storage room. Situated in a boutique building, this condo provides the perfect blend of privacy and proximity to urban amenities. Easy access to public transportation, trendy cafes, and cultural hotspots make this location ideal for the urban dweller.

**Disfavored Description (🧡 AI Realtor, Description B)**: Nestled in the vibrant West Town neighborhood of Chicago, this exquisite loft-style condo at 11 N Green St UNIT 4C offers a light-filled sanctuary that merges modern living with trendy ambiance. Boasting an impressive 1,200 sqft of space, it features expansive windows framing a stunning cityscape backdrop, high ceilings, and an open floor plan, complemented by sleek hardwood floors and a cozy wood-burning fireplace in the living room. Residents will relish the rare luxury of a personal rooftop oasis—ideal for chic entertaining or serene reflection. Positioned among the top 2% for amenities in Chicago, this condo includes in-unit laundry, an intercom system, garage parking, and a large storage room, making it both convenient and stylish. Priced at $490,000, this move-in-ready home offers exceptional value with its central location and effortless access to public transportation, ideal for those seeking the perfect blend of urban convenience and sophisticated living. With its prime spot in Cook County and top-rated area amenities, this condo won't just be a place to live but a lifestyle to love!

**User Comment:** Description B says it is in the top 2% of amenities. What does that even mean. That is nonsense.

## A.2 THE FAILING CASES OF HUMAN-WRITTEN DESCRIPTIONS

We also found cases where users dislike human-written descriptions over model-generated descriptions. The rationales behind the user preferences demonstrate the advantage of model-generated descriptions.

**Disfavored Description (Human, Description A)**: Don't miss out on this sunny 1 bedroom, 1 bathroom condo in the heart of Lincoln Park. Step in to the living space flooded with natural light from your private balcony windows. The open kitchen layout offers tons of storage space, updated stainless steel appliances, and a comfortable breakfast island. In your queen sized bedroom you'll find nicely appointed closets, and cozy carpeting. This perfect starter home features hardwood flooring throughout the living space, and a newer (2021) washer and dryer in unit. Storage included.

**Preferred Description (🧡 AI Realtor, Description B)**: Welcome to 2220 N Sedgwick St APT 204, a charming one-bedroom condo nestled in the vibrant heart of Lincoln Park, Chicago's most coveted neighborhood. Priced competitively at $310,000, this move-in-ready gem offers a harmonious blend of modern convenience and timeless elegance. Revel in the top-tier stainless steel appliances and sun-drenched open kitchen with a cozy breakfast island perfect for any culinary enthusiast. Rich hardwood flooring and cozy carpeting seamlessly blend aesthetics and comfort, while the queen-sized bedroom boasts ample storage. Enjoy serene moments on your private balcony overlooking the iconic neighborhood streets studded with historical charm. Built in 1900, this meticulously maintained brick structure highlights both character and longevity, ensuring a sound investment. With unparalleled access to public transportation and a safe, walkable community, this property meets the highest standards of city living. Enviably situated among Lincoln Park's garden-filled avenues, it's the ideal starter home for those who value location and convenience without compromising on style or safety.

**User Comment:** Description B doesnt even have the size, location, or other important facts.

**Case Analysis:** Sometimes human descriptions even miss important facts, while descriptions generated by our models do not. We present a fine-grained fact-checking study to check whether there is a hallucination in § 5.3.

**Preferred Description (🤖 AI Realtor):** Welcome to 832 W Wrightwood Ave #3, an enchanting 2-bedroom, 1-bathroom condo nestled in the heart of Lincoln Park, Chicago's most prestigious neighborhood. Priced sensibly at $450,000 and boasting a spacious 1,164 sqft of elegant living, this East Lincoln Park penthouse marries historical charm with contemporary amenities. Step inside to discover a warm ambiance highlighted by exposed brick, hardwood floors, and a cozy wood-burning fireplace. The remodeled eat-in island kitchen is an entertainer's dream, seamlessly flowing into a separate dining area perfect for intimate gatherings. With its skylight windows and bay windows, an abundance of natural light illuminates every corner. Enjoy the convenience of an in-unit laundry room, additional private storage, and central air without the high HOA fees typically found in comparable homes. The condo's prime location offers walkability to the vibrant amenities and serene lakefront of Lincoln Park, catering to every lifestyle need. A rare find in a top-tier location with superior accessibility and neighborhood charm, this condo promises both investment value and a delightful urban retreat. Don't miss the open house to experience this gem first-hand!

**Disfavored Description (Human):** WALK TO IT ALL!! THIS BRIGHT TWO BEDROOM, 1 BATHROOM EAST LINCOLN PARK PENTHOUSE W/DECK HAS EXPOSED BRICK, BAY WINDOWS AND A WOOD BURNING FIREPLACE;EAT-IN ISLAND KITCHEN OPENS TO MASSIVE 23' WIDE LIVING ROOM WITH A SEPARATE DINING AREA. THE UNIT HAS BEAUTIFUL HARDWOOD FLOORS THROUGHOUT, A HUGE MASTER SUITE WITH TONS OF CLOSET/STORAGE SPACE. OTHER FEATURES INCLUDE ADDITIONAL PRIVATE STORAGE, IN-UNIT LAUNDRY ROOM WITH SIDE BY SIDE W/D AND PARKING. KITCHEN REMODELED IN 2016, BATHROOM REMODELED IN 2020. NEW AC CONDENSER IN 2022.

**User Comment:** I think this description is much better because it isn't in all caps, which feels like I'm getting yelled at.

**Case Analysis:** Human-drafted descriptions can look unpleasant.

## A.3 THE DICHOTOMY OF USER PREFERENCES ON WRITING STYLES

In § 5.1, we present the aggregated benchmark results to compare the persuasiveness of listing descriptions generated by different models. To get more qualitative insights into the strengths and weaknesses of different models, as well as the subjective nature of human feedback, we present a more detailed case study here.

The first thing we noticed is the users' subtle preferences in **description length**: while some users like concise descriptions that directly go to the point, other users prefer longer descriptions because they want to know more details about the property they are interested. The following two examples of user feedback explain this point.

**Preferred Description (Vanilla, Description A):** Welcome to your dream condo at 4345 S Indiana Ave UNIT 2N, nestled in the vibrant Bronzeville neighborhood of Chicago, IL. This exquisite 3-bedroom, 2-bath home offers 1,550 sqft of modern living infused with classic charm, all for an unbeatable price of $275,000. Built in 2006, it features abundant natural light flooding through large windows, complemented by tall ceilings and an open living space. Imagine cozy evenings by the custom stone wood-burning fireplace or enjoying a morning coffee on your private second balcony. The master bedroom offers tranquility with a spacious walk-in closet, while the additional bedrooms provide generous space for family or guests. The kitchen is a chef's delight, equipped with stainless steel appliances including a range, microwave, and refrigerator. With central air cooling, hardwood flooring, and a sleek, contemporary style highlighted by recessed lighting, this condo is the perfect blend of comfort and sophistication. Adding to the allure, a

secure garage parking spot is included. Security is assured with a modern security system, and the convenience of in-unit laundry completes this superb offering. Located in Cook County with easy access to all Chicago has to offer, this stylish condo is a must-see!

**Disfavored Description (🧑🏽 AI Realtor, Description B):** Welcome to your dream home at 4345 S Indiana Ave UNIT 2N, nestled in the heart of the vibrant Bronzeville neighborhood in Chicago. This stunning condo offers the epitome of comfortable living with 3 spacious bedrooms, 2 modern bathrooms, and a living area of 1,550 square feet, perfectly situated for a single mother seeking convenience and safety. The residence exudes warmth, featuring abundant natural light through large windows and a cozy custom stone wood-burning fireplace in an open living setting. The condo is a gem within the community, boasting one of the top amenities packages in the area, including a stylish stainless steel kitchen, a rare second private balcony, and garage parking that ensures convenience. Step into the master bedroom for a touch of luxury, indulge in the modern ambiance provided by recessed lighting, or relax in the welcoming family room with its captivating atmosphere. Temperature comfort is assured through efficient central air and heating. Notably, this property towers above others in terms of walkability and neighborhood amenities, making it an ideal choice for a family-focused lifestyle. Priced attractively at $275,000, it's a golden opportunity to secure a versatile home that evolves with your needs, ready to create cherished family memories. Discover the potential for a fulfilling life in a community known for its top-tier safety and accessibility, all while investing in a property you can pass down to the next generation.

**User Comment:** Description A gets to the point faster, while still highlighting the important qualities of the home.

**Case Analysis:** Some users love concise descriptions.

---

**Preferred Description (Vanilla):** Welcome to 4454 S Shields Ave, a charming A-Frame single-family home nestled in the heart of Chicago's historic Fuller Park neighborhood. This inviting residence offers three cozy bedrooms and a well-appointed bathroom, all within a compact 956 square feet of open-concept living space that seamlessly combines comfort and style. Built in 1929, the home exudes classic character while featuring modern conveniences such as central air for cooling and a natural gas heating system. The property's allure is further enhanced by its unfinished basement, offering potential for personalized expansions. Imagine summer barbecues on your porch or taking a quick stroll to a nearby park, making this an ideal location for outdoor enthusiasts. With its proximity to local amenities and an incredible price of just $219,900, this home represents a fantastic investment opportunity, especially with its rare, close-to-an-Olympic-sized swimming pool bonus. Discover the potential of this foreclosure property and make it your own urban oasis in Cook County.

**Disfavored Description (SFT):** Welcome to this charming single-family home nestled in Fuller park! This listing features an open concept, 3 bedrooms, 1 full bathroom, and an unfinished basement that's just waiting for your personal touch. Located close to a park with an Olympic-sized swimming pool, you'll have endless recreational opportunities at your doorstep. With its prime location and potential for expansion, this property is a true gem waiting to be polished. Don't miss the chance to make this house your dream home!

**User Comment:** Again, more description is better if I am really interested in a property.

**Case Analysis:** Some users love longer descriptions.

---

Another important factor is the **embellishment** of descriptions. That is, in our particular marketing domain, is there a clear preference towards the embellished or plain style of descriptions. Here are two examples that showcase the different preferences from users:

**Preferred Description (🧑🏽 AI Realtor, Description A):** Welcome to your charming retreat at 7401 S Blackstone Ave, nestled in the vibrant Grand Crossing neighborhood of Chicago! Priced attractively at $179,900, this delightful single-family home skillfully blends affordability and comfort. With three spacious bedrooms and two well-appointed baths, it perfectly accommodates

family living. Highlights of this property include its exceptional flooring, ranking among the top 2% in the area, and the effective recent improvements that make it standout in property condition. Inside, enjoy the luxury of new carpeting and stylish white shaker cabinetry, promising an inviting and modern ambiance. A versatile layout offers flexible options for home customization, meeting the varied needs of your lifestyle, from working to exercising at home. Step outside to a fenced-in yard, ideal for both privacy and play, further increasing its appeal. Though without a garage, the home features machine hookups, ample storage potential in a sub-basement, and a premium location providing easy access to all the amenities and family-friendly spots Chicago's Grand Crossing has to offer. Enhanced by durable brick construction, this home represents a significant investment opportunity, combining all the elements you desire in a new haven. Your perfect blend of convenience, comfort, and charm awaits!

**Disfavored Description (Vanilla, Description B)**: Welcome to 7401 S Blackstone Ave, an inviting single-family home nestled in Chicago's charming Grand Crossing neighborhood. Priced attractively at $179,900, this 3-bedroom, 2-bathroom gem is perfect for those seeking comfort and convenience in an established community. Built in 1973, the residence boasts 933 square feet of living space and is beautifully updated with new carpeting throughout and stylish newer flooring. The kitchen shines with modern white shaker cabinetry, providing a fresh, contemporary feel. The home is well-equipped for practicality, featuring a dedicated laundry room with machine hookups for both gas and electric dryers. Adding to its appeal is a sub basement and a fenced-in yard, creating an ideal outdoor space for families or pet owners to enjoy. Conveniently located in Cook County, this home is serviced by Lake Michigan water and public sewer, and its brick construction ensures durability. With natural gas and forced air heating, you'll be cozy year-round. This delightful abode represents a fantastic opportunity for homeownership without the burden of HOA fees. Don't miss your chance to make this delightful Chicago residence your own!

**User Comment:** Description A is a bit more descriptive without going overboard, also talks about the neighborhood.

**Case Analysis:** Some users love more ==descriptive== descriptions.

**Disfavored Description (🧌 `AI Realtor`, Description A):** Nestled in the heart of Chicago's vibrant Bridgeport neighborhood, 3457 S Lituanica Ave offers unparalleled access and convenience, situated comfortably within Cook County. This spacious five-bedroom, two-bathroom single-family home is a standout choice, boasting top-tier features in location, accessibility, and outdoor living spaces. With its robust brick construction, this property provides a durable and inviting home environment, perfect for customization to suit your family's evolving needs. Enjoy the luxury of a generous 6,500 sqft lot, among the best in its zipcode, offering a blank canvas for your dream garden or a secure playground for your child. The home's interior shines with elegant hardwood flooring and practical features like in-unit laundry with sink. Practical comfort is ensured with space pac cooling and efficient natural gas heating, ensuring you feel at home year-round. Embrace Chicago living with easy access to nearby amenities, public transportation, and renowned neighborhood characteristics, all for an attractive price point of $549,000—making it an excellent investment for future growth.

**Preferred Description (Vanilla, Description B):** Welcome to your future home at 3457 S Lituanica Ave, nestled in the heart of Chicago's vibrant Bridgeport neighborhood. This charming single-family residence offers five spacious bedrooms and two full bathrooms, perfect for families seeking both comfort and style. Priced at an attractive $549,000, this home sits on a generous 6,500 sqft lot, providing ample outdoor space for relaxation or entertaining. Crafted with enduring brick construction, the property boasts modern conveniences including a complete suite of appliances like a range, microwave, dishwasher, and more. The elegant hardwood flooring throughout adds a touch of sophistication, while the first-floor full bath caters to easy accessibility. Enjoy the convenience of in-unit laundry with a dedicated sink and stride out onto your private deck for a breath of fresh air. The two-car garage offers security and storage, supported by reliable utilities such as public sewer, natural gas heating, and Space Pac cooling. With easy access to Holden Elementary and local amenities, this home represents a delightful blend of classic charm and modern living in one of Cook County's most desirable neighborhoods. Don't miss the opportunity to make this house your home.

**User Comment:** Description B does a better job at listing the amenities.

**Case Analysis:** Some users love a ==plain style== of description that listing all amenities.

These obervations suggest that there is no one-size-fits-all solution for writing style. Hence, future work could consider tailoring the description generation in the user's preferred writing style to further improve the persuasiveness.

## A.4 THE DIVERSITY OF WRITING STYLES ON DIFFERENT LISTINGS

🧌 `AI Realtor` shows diverse writing styles linguistically on listings based on their different features, which means it can tailor different real estate listings well.

In the following pair of examples, Low-end listings emphasize "Safety & Survival" (security, enclosure, reassurance), whereas high-end listings emphasize "Display & Views" (openness, visual richness, and mastery over the environment).

**Low-Price Representative ($110,000).**

> **Focus: Defense, Enclosure, Reassurance**. Words aimed at eliminating buyer insecurity regarding the environment.
>
> **$110,000, 750.0 sqft, 2 beds, 1 bath**
>
> Welcome to your charming oasis at [address], nestled in the vibrant and culturally rich Hyde Park neighborhood of Chicago. This inviting 2-bedroom, 1-bathroom condo offers the perfect blend of comfort and convenience at an unbeatable price of $110,000. Step inside to discover a sun-drenched living space adorned with hardwood flooring and an updated kitchen featuring modern appliances, including a wine refrigerator. The thoughtful design includes first-floor conveniences like a full bath and ample storage, with walk-in closets providing plenty of room for your essentials. Enjoy **tranquil moments** on the large back deck, ideal for relaxation or entertaining guests, set within a **gated courtyard that ensures privacy and security**. The property is **meticulously maintained**, boasting brick construction and a welcoming community atmosphere. Although it is **compact**, the space is optimized for comfortable living **without unnecessary upkeep**, perfect for those valuing efficiency. With **proactive security measures**, a **strong sense of community**, and only minutes away from necessities, this condo perfectly encapsulates the ideal home for those **prioritizing safety** and cultural alignment in a vibrant neighborhood.

**High-Price Representative ($1,875,000).**

> **Focus: Aggression, Openness, Visuals**. Words aimed at showing off transparency and mastery over the environment.
>
> **$1,875,000, N/A sqft, 4 beds, 4 baths**
>
> Discover **unparalleled elegance** and style at [address], a single-family haven nestled in the vibrant Bucktown neighborhood of Chicago. This **exquisite home**, priced at $1,875,000, offers four bedrooms and four bathrooms, perfect for families seeking **ample space and luxury**. Its standout features include a **private corner lot** and a **spacious side yard** designed for **ultimate outdoor enjoyment**, complemented by **gourmet enhancements** like a custom kitchen and a chic beverage center. New Pella windows and **cascading expanses of glass** invite an **abundance of natural light**, creating a bright and airy atmosphere across a versatile loft area ideal for work-from-home needs. With **sophisticated enhancements** such as vaulted ceilings, multiple fireplaces, and a gas fire table, this residence exudes comfort and warmth year-round. The **meticulously crafted design** places this property among the **top tier in architectural style** and elegance within the neighborhood and beyond. Enjoy **seamless access** to essential amenities and natural beauty, with a spacious parking capacity for four cars. **Embrace this rare opportunity** to own a piece of **refined luxury** in an urban yet serene setting.

## B    THE DESIGN OF SURVEY AND USER INTERFACES

### B.1    SURVEY SCREENING INTERFACE

The first stage of the survey is designed to ensure the human subject has sufficient experience in the home search process in order to analyze the features from a marketing description. We present description of an example listing and design quiz-like questions to verify whether the participant is able to make all correct responses. We showcases the web user interfaces in Figure 6.

### B.2    PREFERENCE ELICITATION INTERFACE

In the second stage of the survey, we design an interface to mimic the environment of online platforms that the model can observe the buyer's general profile and behaviors (e.g., recently browsed or liked listing) to some degree. In our case of real estate listing, we ask the buyer to provide their preferences in a 1-5 scale on five general categories (price, location, home features & amenities, house size, investment value) and set a filter on the price range and number of bedrooms in the house they are looking for. This information allows us to select generally relevant listings to mitigate the anchoring effect that the marketing content can play little role to influence the buyer in the

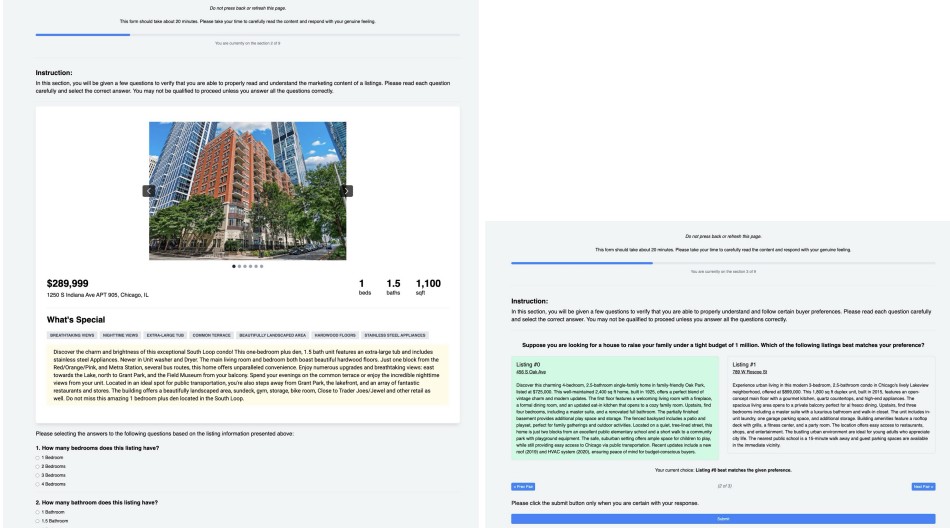

Figure 6: Survey Screening Interface

evaluation phase. Next, we choose 5 relevant listings and ask the buyer to rate them on a 1-5 scale and provide their reasoning. This process ensures that we can collect a reasonable amount of each buyer's preference information for the personalized persuasive content generation in the evaluation phase. Finally, we employ LLM to narrow the features that are likely preferred by the participants and ask for their ratings of importance on a 1-5 scale. We showcases the web user interfaces in Figure 7.

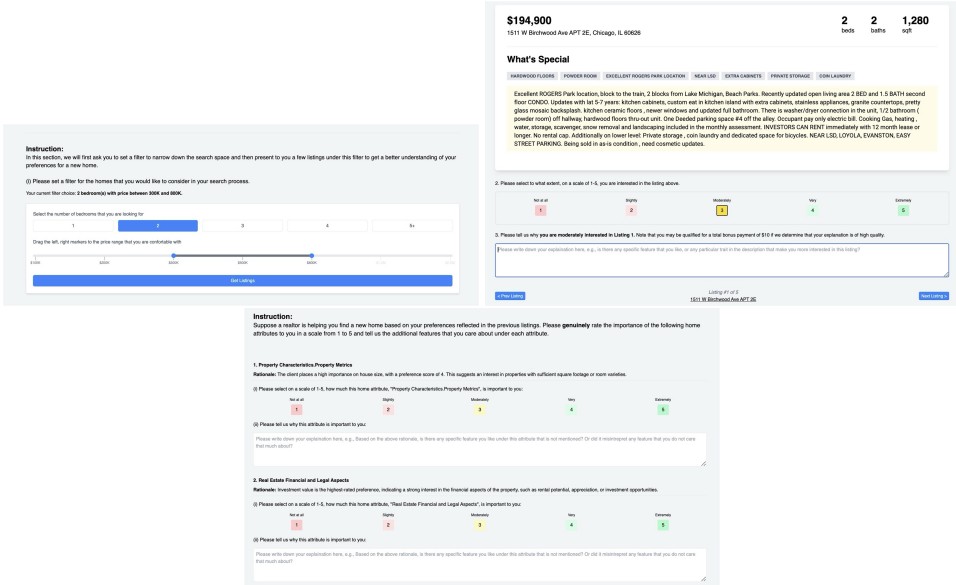

Figure 7: Preference Elicitation Interface

### B.3 HUMAN EVALUATION INTERFACE

In the last stage of the survey, it is to gather the human feedback on the persuasiveness of different models. Many previous works study persuasion by asking human how much does their opinion changes before and after reading an argument. In our task, human subjects often do not have any prior knowledge about item and this evaluation procedure would induce bias. Instead, we implement two alternative evaluation schemes in our interface: one is the A/B test where the buyer is presented

with a single listing along with two descriptions generated by two distinct models and then asked to report which description makes them more interested in the listing; the other is the interleaved test where a set of listings each with a single description generated by some model and the buyer is asked to select the listings that they are interested in based on their descriptions. Each time after a participant's choice of the preferred description, we ask participant to rate on a scale of 1-5 that one description is prefer over another and incentivized them to provide a detailed rationale of their responses. To illustrate this process, we present the web interface design in Figure 8.

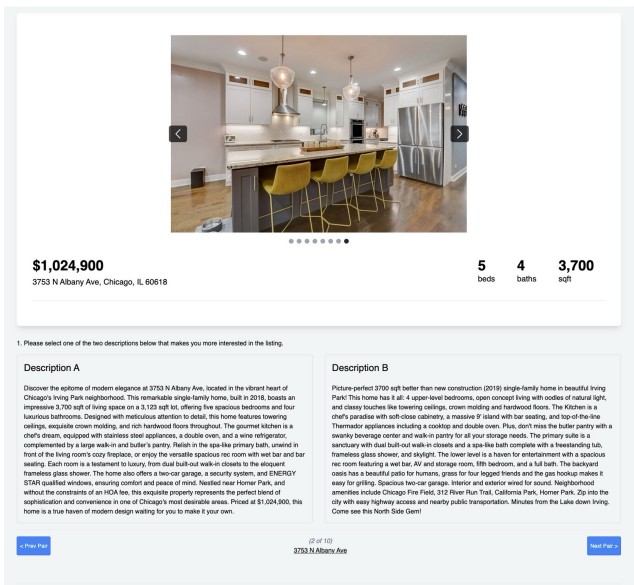

Figure 8: Human Evaluation Interface

### B.4 Feature Annotation Interface

To ease the task of feature annotation, we also develop a user-friendly web interface. Its design is shown in Figure 9.

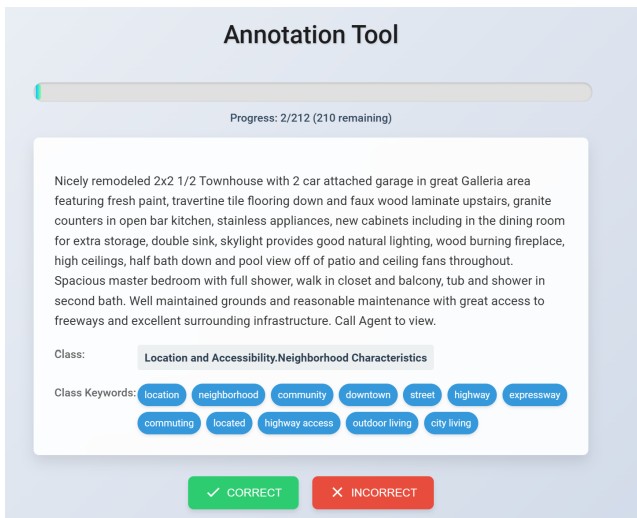

Figure 9: Annotation Interface

# C    IMPLEMENTATION DETAILS

In this section, we provide a full description of the implementation detail of 🧸 AI Realtor.

## C.1    GROUNDING MODULE: PREDICTING MARKETABLE FEATURES

Our model assumes the existence of attribute-feature mappings in different marketing problems, with which a seller can use to influence the buyer's beliefs and behaviors. However, a key challenge lies in determining how to accurately obtain such mappings. Specifically, we must identify which *signaling features* to include and under what conditions it is natural to market a product as possessing a particular feature. Traditionally, acquiring this knowledge from human experts is both labor-intensive and costly. Instead, we take a learning approach to uncover the mapping from our experiment dataset. While the raw dataset contains no annotation of any signaling feature, we employ LLMs to construct a high-quality feature schema and label the dataset accordingly in preparation for learning the attribute-feature mapping. This approach notably presents a novel unsupervised learning paradigm, harnessing the broad knowledge of LLMs to distill expert-level insights from unlabeled data with minimal human supervision.

**Inductive Construction of Feature Schema**  Our dataset only contains the raw attributes of each product. In order to learn a high-quality attribute-feature mapping, the first task is to obtain a good representation of feature schema $S$. On the one hand, if we miss some useful signaling features, it could significantly hinder the performance of subsequent marketing task. On the other hand, there are so many possible token that can serve as the signaling features in the natural language space, and many of these tokens might have duplicate or similar meaning. If there is no structured representation of the features, the resulting label classes could be too sparse to learn. Indeed, we discover that the feature schema obtained by directly prompting an LLM includes many similar features while miss some important ones. Based on this observation, we turn to a more sophisticated prompting strategy to inductively improve the quality and representation of the feature schema (see a high-level sketch of the construction pipeline in Figure 2).

First, we construct a basis of feature schema, represented as a list of tokens used in the human-written marketing description to describe some house features. We begin with *Mixtral-8x7B-Instruct-v0.1* (Jiang et al., 2024) to extract keywords or phrases $\{k_1, k_2, \dots\} = \text{LLM}_{\text{gen}}([\mathcal{I}_{\text{Keyword}}; D_{\text{human}}])$ that summarize each human-written description $D_{\text{human}}$ under a keyword-extraction prompt $\mathcal{I}_{\text{Keyword}}$ (Appendix F.1). We observed that, in some cases, the model output could not be directly parsed into a clean list of keywords, or it contained excessive quantifiers and modifiers. To address this, we re-prompted the model using $\mathcal{I}_{\text{Norm}}$ (Appendix F.2) to normalize each keyword. Through this process, we initially extracted 112688 keywords—too many to handle effectively. We then applied additional normalization steps, including lowercasing, lemmatization, and synset merging via NLTK (Bird et al., 2009). We also filtered the keywords, retaining only those that appeared in at least 50 descriptions. This reduced the final set to 1114 keywords as our *induction base*.

Next, we organize the feature-related keywords into a structured feature schema. Since many keywords are related to each others and hard to distinguish, we use a hierarchical representation of feature schema to better capture the relations between different feature classes and to ease the subsequent labeling task. To achieve this goal, we prompted *Claude-3.5-Sonnet* (Anthropic, 2024) with a 100-keyword batch to iteratively generate a hierarchical schema that covers the majority of the keywords (an example run can be found in Appendix F.3). We temporarily switched to *Claude-3.5-Sonnet* because we found it particularly difficult for open-source models, even the state-of-the-art *GPT-4o* (OpenAI, 2024a), to induce such a schema without grouping most keywords into overly broad categories like "others" or "misc", resulting in a shallow and uninformative schema. In contrast, when fed keywords in small batches, *Claude-3.5-Sonnet* followed our instructions more faithfully, organizing the keywords into a carefully structured hierarchy. Every leaf node in the schema was associated with a set of relevant keywords. From this process, we obtain a relatively well-structured and comprehensive feature schema.

Finally, to evaluate the quality of the generated feature schema, monitor potential hallucination issues, and further refine the schema, we asked three human participants to conduct manual review. We prompt *Mixtral-8x7B-Instruct-v0.1* to determine whether a feature from the schema presents in each human-written description, and each participant is asked to independently verify this result (see

our annotation interface in Appendix B.4). Based on the participants' feedback on 636 samples, we found that features labeled by LLMs are mostly agreed across all human annotators, except for some ambiguous or subjective features (e.g., the aesthetic features of a house), where the agreement rates (around $60\%$) between models and human are about as good as that among human annotators. We refine the schema for two more iterations, where we prompt LLMs to merge some similar features and reduce the ambiguity of some features with more precise example keywords. We list our final feature schema in Appendix D.2 and it is used in the subsequent stages of our pipeline.

**Learning the Feature-Attribute Mapping**  With the feature schema, we guide the LLM to annotate for each product with attributes $\mathbf{x}$ whether each feature $s_i$ is described in the human-written marketing text (see the prompt in Appendix F.4). We perform a few additional pre-processing steps to this correspondence data to supervise the learning of the feature-attribute mapping.

First, we found that some human-written marketing descriptions are of relatively low quality and these data points can negatively impact the learnt feature-attribute mapping. Hence, we only select marketing descriptions of products that are relatively popular, according to a simple heuristic ratio between the number of likes and views received by a listing recorded on the marketing platform. We expect the quality of feature-attribute mapping uncovered from this filtered set of human-written descriptions would be higher than average.

Next, we normalize the attributes of each listing $\mathbf{x}$ and embed existing knowledge of these attributes into their representation. Since the raw attributes of each listing $\mathbf{x}$ have different value types (categorical, integer, float, etc.), we convert each attribute $x_i$ into a natural language statement using the template, "The attribute *attribute_name* is *attribute_value*.", and then use an embedding model, *SFR-Embedding-Mistral* (Meng et al., 2024), to convert each natural language statement into a fixed-dimensional vector $e_i = \text{LLM}_{\text{embed}}(x_i) \in \mathcal{R}^d$. We also perform some standardized normalization techniques such as removing irrelevant attributes and dropping attributes with missing values. Finally, we use a simple multi-layer perceptron (MLP) to learn the attribute-feature mapping as,

$$\pi(s_i \mid \mathbf{x}) = \sigma(O_i^T \text{ReLU}(W \bar{e}(\mathbf{x}))),$$

where $\bar{e}(X)$ is the mean-pooled attribute embedding, and $O_i \in \mathcal{R}^{d/2}, W \in \mathcal{R}^{d \times d/2}$ are the model's weights. The function $\sigma$ represents the sigmoid activation function. Here, we assume conditional independence between highlights given the raw features $X$. We use the standard logistic loss function to training the neural network. We apply a random train-test split of $4 : 1$ ratio in our dataset and achieve testing accuracy $69.39\%$ and F1 score $67.43\%$. We find the accuracy to be reasonably high, given the stochastic nature of signaling process. That is, the features deterministically predicted based on our mapping cannot exactly match with the features used in the human written description with some degree of randomness — just as the accuracy of predicting a fair coin toss is at most $50\%$.

The typical implementation of a signaling scheme is to follow the attribute-feature mapping $\pi$ to randomly draw a signal $S_j$ with probability $s_j(\mathbf{x})$. This is necessary in theory to maintain the partial information carried by each signal. However, we implement a deterministic feature selection strategy to only use feature $S_j$ with probability above some threshold $\alpha$. This is because our generated marketing content only accounts for a tiny portion of the corpus so that it should have almost no influence on people's perception of a feature (e.g., the partial knowledge inferred upon observing each feature). This also ensures that the product would have the feature with high probability, as our objective prioritizes the rigorousness of our marketing content. As a simple heuristics in our implementation, we set the threshold $\alpha = 1/2$ and we will refer to this set of features as,

$$\text{Marketable Features:} \quad \mathcal{S}_1(\mathbf{x}) = \{S_j : s_j(\mathbf{x}) \geq \alpha\}. \tag{3}$$

## C.2 PERSONALIZATION MODULE: ALIGNING WITH PREFERENCES

This stage seeks to steer the persuasive language generation toward the buyer's preference, which is another crucial objective of grounded persuasion. In particular, with the advent of LLM, there is an unprecedented opportunity for our data-driven approach could achieve much higher degree of personalization with significantly lower cost than the conventional marketing designed for a larger population. Our solution has two parts: the first part is to properly elicit the useful information about a user's preference and structure it in a good representation; the second part is to select a subset of features based on the user preference in order to maximize the influence to the user's belief.

**Structured Preference Representation**  As mentioned previously, our evaluation environment is built to have an information elicitation process from each buyer. However, such information cannot directly describe the user's preference. So, we ask the LLM to act like a human realtor to determine the features that the users might be interested in based on their initial selection. To do this, we prompt the language model to convert the user preference into information structured according to the feature schema. We then ask the user to give a rating $r_j$ on a scale of 1-5 on how important each feature $S_j$ is. We also elicit the user's rationale behind this rating to nudge users to give more thoughts on their selection and thereby improve the credibility of their rating responses. While our implementation mostly relies on user surveys and the information processing power of LLMs, this design is a reasonable simulation of digital marketing in real-world applications, where $r_j$ can be learned through the standard industrial techniques of cookie analysis.

**Personalized Feature Selection**  While the marketable features in Equation (3) are predicted at a population level, it is also useful to select features that are tailored to the user's special interests. However, because real-world marketing descriptions are not optimized for individual users, we cannot simply rely on a data-driven machine learning approach for personalization. Instead, we leverage the innate capability of LLMs to understand and analyze human preference. In our implementation, we select a set of features that are marketable and preferred by the buyer and let the LLMs to decide which personalized features to emphasize on in the marketing content. Our heuristic method for personalized feature selection is to adjust the population-level feature scores $\mathbf{s}(\mathbf{x})$ with the user's rating over each feature $\mathbf{r}$ as follows,

$$\text{Personalized Features: } \mathcal{S}_2(\mathbf{x}) = \{s_j | s_j(\mathbf{x}) + c(r_j - r_0) \geq \alpha\}, \tag{4}$$

where the constant $c$ reflects the intensity of personal preference, $r_0$ is the basis rating of each attribute. In our human-subject experiment, we choose $c = 0.01$, $r_0 = 2$ and set the threshold value $\alpha$ such as to select features of the top 10 highest scores. We list these features in the prompt to generate persuasive marketing description (see a full specification in Appendix F.5).

### C.3 MARKETING MODULE: CAPTURING SURPRISAL VIA RAG

The last stage is designed to better ground the persuasive language generation on factual evidences, problem contexts and localized information in automated marketing. There are many ways to improve the grounding for different settings of automated marketing. As a case study, we choose to focus on the surprising effect, a common marketing strategy studied by many work (Lindgreen & Vanhamme, 2005; Ludden et al., 2008; Ely et al., 2015), under which the buyers would derive entertainment utility and have a deeper impression. In our setting of real estate marketing, we consider the type of features that are relatively rare in its surrounding area. That is, we say a marketable feature $S_j$ is *surprising* if it is among the top $\beta$-quantile of the distribution of $S_j$ values under the prior distribution $s_j(\mu)$, or formally,

$$\text{Surprising Features: } \mathcal{S}_3(\mathbf{x}) = \{S_j \subset \mathcal{S}_1 :$$
$$s_j(\mathbf{x}) \text{ is within } \beta\text{-quantile of distribution } s_j(\mu)\}. \tag{5}$$

In our implementation, we determine a set of features for each listing that have its comparative advantage among different groups of similar listings. We consider two kinds of retrieval criteria: (1) select all listings within the proximal location at different levels of granularity (e.g., neighbourhood, zipcode or city); (2) select the 10 listings with the most similar features via an information retrieval system (implemented by the ElasticSearch framework[6]) — the search engine implementation details can be found in Appendix F.8. For each group of similar listings, we determine an empirical distribution function on each attribute score $\tilde{F}_i$. We then set $1 - \tilde{F}_i(p_i)$ as the percentile ranking of the listing's attribute $i$ among this group. We then select all attributes that are among the top $30\%$ percentile ranking for some group and provide the information in the prompt to generate persuasive marketing language (see a full specification in Appendix F.6). This gives the LLMs localized feature information at different granularity level.

---

[6]https://www.elastic.co/elasticsearch

# D DATA CURATION

## D.1 DATASET RAW ATTRIBUTE SCHEMA

To ensure both quality and fidelity of our evaluation, we collect the real data of real estate listings on the market. The dataset for this experiment was sourced primarily from Zillow and includes around 50000 listings collected in the month of April in 2024. We follow the Zillow terms of services[7] to avoid any commercial use of their data. Each of these listings is from one of the top 30 most populous cities in the United States as described by the U.S. Census Bureau. Listings that were not residential in nature or were missing crucial data to this experiment were excluded from this dataset. This dataset is composed of 95 columns, with features ranging from number of bedrooms, price, views, and more (see Table 1). These many features associated with each listing provide us sufficient space to develop and test improved models for grounded persuasion.

| Field Name | Data Type |
|---|---|
| bedrooms | float64 |
| bathrooms | float64 |
| price | float64 |
| description | object |
| living_area_value | float64 |
| lot_area_value | float64 |
| area_units | object |
| brokerage_name | object |
| zipcode | object |
| street_address | object |
| home_type | object |
| time_on_zillow | object |
| page_view_count | float64 |
| favorite_count | float64 |
| home_insights | object |
| neighborhood_region | object |
| scraped_at | object |
| url | object |
| city | object |
| state | object |
| year_built | float64 |
| county | object |
| avg_school_rating | float64 |
| id | object |
| time_on_zillow_days | float64 |
| score | float64 |
| jpeg_urls | array |

Table 1: Listing data, subset of important columns

## D.2 FINAL FEATURE SCHEMA

Here is the condensed version of the final feature schema to save pages:

```
Interior Features:
    Rooms:
        [bath,bathroom,bedroom,kitchen,living room,secondary
            ↪ bedrooms,patio,backyard,closet,room,living,dining
            ↪ room,pantry,space,office,laundry room,dining,living
            ↪ space,living area,primary suite,master suite,family
            ↪ room,cellar,foyer,game room,great room,den,master
```

---

[7]https://www.zillow.com/z/corp/terms/

```
          ↪ bedroom,utility room,sunroom,bedroom suite,living
          ↪ areas,primary bedroom,office space,kitchenette,owner
          ↪ 's suite,playroom,storage room,living rooms,ensuite,
          ↪ wet bar,loft area,sitting room,mud room,exercise
          ↪ room,clothes closets,walk-in closet,mudroom,
          ↪ conference room]
    Flooring:
        [flooring,stories,carpeting,hardwood floors,tile,tile
          ↪ floors,hardwood flooring,wood flooring,hardwood
          ↪ floors]
    Furniture:
        [desk,table,chair,bed,dressers,cupboards,sofa,bench,
          ↪ seating]
    Additional Spaces and Versatility:
        [bonus room,flex space,flex room,den]
    Kitchen Features:
        [countertop,granite countertops,marble countertops,island,
          ↪ cabinetry,kitchen island,kitchen cabinets,waterfall,
          ↪ dining space,cooktop]
    Architectural Elements:
        [roof,window,floor plan,cabinet,molding,staircase,brick,
          ↪ paneling,siding,beam,ceiling fans,stair,chandelier,
          ↪ finishing trim,baseboard,trim]
    Bathroom Features:
        [shower,vanity,powder room,jacuzzi,ensuite,half bath,water
          ↪  closet,mirror,faucet]
    Storage:
        [storage,closet space,cabinet space,shelving,storage space
          ↪ ,mudroom,drawer,bookshelf,storage unit,clothes
          ↪ storage,bike storage]
    Comfort and Ambiance:
        Lighting:
            [lighting,natural light,light fixtures,skylight,
              ↪ lighting fixtures]
        Temperature Control:
            [fireplace,hvac,fan,ac,a/c,central air conditioning]
Exterior Features:
    Outdoor Spaces:
        [patio,backyard,yard,pool,spa,balcony,porch,deck,roof deck
          ↪ ,outdoor space,rv parking,outdoor spaces,outdoor
          ↪ living space,fenced yard,pavers,garden,outdoor
          ↪ living,backyard oasis,pergola,gazebo,cabana,
          ↪ landscaping,shade,lawn,fountain,sod,outdoor bench]
    Outdoor Activities:
        [gardening,outdoor cooking,barbecue,bbq]
Location and Accessibility:
    Neighborhood Characteristics:
        [location,neighborhood,community,downtown,street,highway,
          ↪ expressway,commuting,located,highway access,outdoor
          ↪ living,city living]
    Nearby Amenities:
        [shopping,restaurant,park,school,grocery,cafe,hospital,
          ↪ food,stadium,museum,boutique,shopping centers,
          ↪ station,elementary,bus,trader joe's,golf,brewery,
          ↪ elementary school,school district,recreation
          ↪ facility]
    Cities/Regions:
```

```
        [Austin,Denver,Charlotte,Houston,Dallas,San Antonio,
        ↪ Nashville,Phoenix,Los Angeles,LA,Manhattan,Detroit,
        ↪ Philadelphia,Portland]
    Access and Transportation:
        [access to amenities,proximity to schools,proximity to
        ↪ restaurants,proximity to shops,access to shopping,
        ↪ bus stop,walking distance,proximity to shopping,
        ↪ freeway access,public transit nearby,public
        ↪ transportation,road]
    Walkability and Bikeability:
        [walkability,bike score,walk score]
Housing Types:
    [studio,cottage,ranch,duplex,townhome,brownstone,row home,
    ↪ bungalow]
Building Features:
    Structure:
        [condo,loft,unit,townhouse,estate,square feet,duplex,
        ↪ garage,carport,story,penthouse,sf,triplex,colonial]
    Parking:
        [garage,parking,parking space,parking spaces,garage door,
        ↪ parking spot]
Appliances:
    [appliance,refrigerator,dishwasher,washer/dryer,range,fridge,
    ↪ microwave,washer,ac unit,dryer,hood,laundry facilities,
    ↪ washer and dryer,oven,garbage disposal,wolf appliances,
    ↪ thermador appliances]
Amenities:
    [community center,community pool,spa,firepit,fire pit,
    ↪ outbuilding,tennis courts,club house,rooftop,rooftop
    ↪ deck,rooftop terrace,dog park,lounge,elevator,recreation
    ↪  room,gym,fitness center,clubhouse,swimming pool,pool,
    ↪ spa,sauna,hot tub,putting green,tennis courts,basketball
    ↪ ,pickleball,tennis court,golf,management,booking,
    ↪ concierge,trash,maintenance,doorman,superintendent,
    ↪ nightlife,brewery]
Utilities and Systems:
    [plumbing,water heater,heater,hot water heater,water,water
    ↪ filtration system,gas,sprinkler system,hvac,ac,a/c,
    ↪ wiring,solar panels,solar,electrical panel,electricity,
    ↪ generator,security,security system,camera,internet,wifi,
    ↪ cable,phone,satellite,fiber,internet access,satellite TV
    ↪ ,internet service,irrigation system,ac unit,hvac unit,
    ↪ central air conditioning]
Design and Style:
    Interior Design:
        [paint,style,home style,architecture,woodwork,ensemble,
        ↪ accent,open floor plan,drawing]
    Aesthetics:
        [elegance,sophistication]
    Architectural Styles:
        [tudor,colonial,craftsman,farmhouse]
Smart Home Features:
    [smart home technology,surround sound,home technology,camera]
Lifestyle Features:
    Work from Home:
        [workspace,home office]
    Entertainment:
        [entertaining space,party,entertainment options,wet bar,
        ↪ entertainment]
```

```
Sustainability Features:
    [solar system,sustainability,solar,heated floors,solar panels,
        ↪ tankless water heater]
Real Estate Financial and Legal Aspects:
    [condo fee,hoa fee,hoa fees,equity,hoa dues,condo fees,cdd
        ↪ fees,occupied,rental potential,income potential,
        ↪ appreciation,airbnb,investment opportunity,investor
        ↪ opportunity,warranty,pricing,rental income,income,
        ↪ financing,utility,sale,closing,furnished,slip,tax,flip
        ↪ tax,abatement,zoning,hoa,rental cap,option]
Water Features:
    [soaking tub,softener]
Views and Scenery:
    [mountain views,lake views,ocean views,sunset,city views,
        ↪ skyline,skyline views]
Property Characteristics:
    Specialty Rooms:
        [wine cellar,media room,suite]
    Distinctive Interior Elements:
        [exposed brick,high ceilings]
    Exterior Appearance:
        [curb appeal,facade,exterior paint]
    Atmosphere:
        [oasis,retreat,sanctuary,flow]
    Environment:
        [surroundings]
    Property Metrics:
        [lot,corner lot,sqft,br,walk score,foot,inch]
    Property Condition:
        Improvements:
            [improvement,tlc,fixer,flooded]
        Age and Status:
            [new,renovated,remodeled,renovated,rehabbed,home age,
                ↪ upgrade,update,built,finish,updated,move,
                ↪ readiness,move-in ready,maintained]
Real Estate Industry:
    [builder,agent]
```

### D.3 DIVERSITY OF THE REAL ESTATE MARKET IN CHICAGO

In this section, we analyze the diversity of the real estate market in Chicago compared to other major US cities. We use two quantitative signals: (1) the diversity of home types measured by entropy, and (2) the dispersion of prices measured by percentile ratios (p90/p10 and p75/p25). Higher values in either metric indicate a more heterogeneous market. The home type entropy for each city is summarized in Table 2, and the cross-city price dispersion is reported in Table 3.

Overall, Chicago emerges as the most diverse market among the major cities examined. As shown in Table 2, it has the highest home type entropy, indicating a well-balanced mix of condos, single-family homes, multi-family units, and townhouses. Meanwhile, Table 3 shows that Chicago also exhibits the strongest price dispersion, reflecting a wide range of housing options across different price tiers. Together, these signals highlight Chicago as a particularly heterogeneous and versatile real estate market.

## E HALLUCINATION EXPERIMENT DETAILS

In this section, we introduce implementation details for hallucination verification experiments. We will introduce both automatic evaluation and human evaluation.

Table 2: Home type entropy across major US cities. Higher entropy indicates a more balanced home-type distribution. Chicago exhibits the highest diversity.

| City | Home Type Entropy |
|------|-------------------|
| **Chicago, IL** | **0.8613** |
| Seattle, WA | 0.8415 |
| San Jose, CA | 0.8399 |
| Los Angeles, CA | 0.8018 |
| San Francisco, CA | 0.7851 |
| Washington, DC | 0.7796 |
| Portland, OR | 0.7434 |
| Denver, CO | 0.7064 |
| San Diego, CA | 0.6849 |
| Philadelphia, PA | 0.6375 |

Table 3: Price dispersion across cities. Higher percentile ratios indicate larger heterogeneity in listing prices. Chicago shows the strongest price dispersion.

| City | Price p90/p10 | Price p75/p25 |
|------|---------------|---------------|
| **Chicago, IL** | **10.09** | **3.36** |
| Seattle, WA | 5.44 | 2.07 |
| San Jose, CA | 4.81 | 2.32 |
| Los Angeles, CA | 5.48 | 2.37 |
| San Francisco, CA | 5.30 | 2.28 |
| Washington, DC | 7.03 | 2.50 |
| Portland, OR | 5.09 | 2.24 |
| Denver, CO | 5.85 | 2.47 |
| San Diego, CA | 5.44 | 2.32 |
| Philadelphia, PA | 5.97 | 2.38 |

### E.1 AUTOMATIC EVALUATION

We adopt fine-grained fact-checking based on GPT-4o for automatic evaluation, similar to the pipeline introduced in FActScore(Min et al., 2023). Specifically, we select *price*, *living area* (in sqft), *#bedrooms* and *#bathroom* as $X_{\text{hard}}$ and *home insights*, *address* as $X_{\text{soft}}$ according to a prior survey of user preference.

We use structured output API[8] on OpenAI to setup $\text{eval}_{\text{soft}}(L, x)$ and $\text{eval}_{\text{hard}}(L, x)$. This means in both cases, we need to first define the structured output class specification and then prompt the model with it.

For Faithful$_{\text{hard}}$, our structured output class specification is:

```
class MainInfo(BaseModel):
    price_mentioned: bool
    price: float
    living_area_mentioned: bool
    living_area: str
    bedrooms_mentioned: bool
    bedrooms: float
    bathrooms_mentioned: bool
    bathrooms: float
    address_mentioned: bool
    address: str
```

and our prompt for $\text{eval}_{\text{hard}}(L, x)$ is:

---

[8]https://platform.openai.com/docs/guides/structured-outputs/introduction

```
messages=[
    {"role": "system", "content": "Extract Real Estate Information
        ↪ . Find the price (e.g, 290000.0), living area (e.g.,
        ↪ '990.0 sqft'), bedrooms (e.g., 2) and bathrooms (e.g.,
        ↪ 3) from the description. Not all information may be
        ↪ present, so you also have to determine whether each
        ↪ field is mentioned or not."},
    {"role": "user", "content": {description}}
]
```

We then compare the extracted information with $\text{supp}(L, X_{\text{hard}})$ to compute Faithful$_{\text{hard}}$. If certain attributes are mentioned (i.e., *xx_mentioned*=True) and the corresponding extracted values matched the listing info $\text{supp}(L, X_{\text{hard}})$, then we will give one score, otherwise zero.

For Faithful$_{\text{soft}}$, we will compute it in two stages. First, we will conduct attribute extraction as in Faithful$_{\text{hard}}$, but with a different set of attributes $X_{\text{soft}}$. Our structured output class specification is:

```
class MainInfo(BaseModel):
    home_insights_mentioned: bool
    home_insights: list[str]
    address_mentioned: bool
    address: str
```

and our prompt is:

```
example_home_insights =["Large island", "Oversized bathroom", "
    ↪ Open floor plan", "Lake views", "Orange l lines", "Newer
    ↪ stainless steel appliances", "Gorgeous hardwood floors", "
    ↪ Tons of cabinet space", "In-unit washer and dryer", "Skyline
    ↪  view", "Private balcony", "Beautiful city"]
example_addr = "1255 S State St UNIT 703 Chicago IL 60601"
messages=[
    {"role": "system", "content": "Extract Real Estate Information
        ↪ . Find the home insights (e.g., {example_home_insights})
        ↪ , and address (e.g., {example_addr}) from the
        ↪ description. Not all information may be present, so you
        ↪ also have to determine whether each field is mentioned
        ↪ or not."},
    {"role": "user", "content": {description}}
]
```

In the second stage, we will use JSON mode API[9] to check whether the extracted attributes match $\text{supp}(L, X_{\text{soft}})$. Our matching prompt is:

```
Given the following information:

1. Description: {description}
2. True value for {attribute_name}: {json.dumps(true_value)}
3. Extracted value for {attribute_name}: {json.dumps(
    ↪ extracted_value)}

Please analyze how well the extracted value matches the true value
    ↪ , considering the context provided in the description.

For 'home_insights', consider it a good match if a significant
    ↪ subset of the true insights is correctly identified.
For 'address', consider it a good match if at least a subset (e.g
    ↪ ., city/state) is correctly identified, given it was
    ↪ mentioned in the description.
```

---

[9]https://platform.openai.com/docs/guides/structured-outputs/json-mode

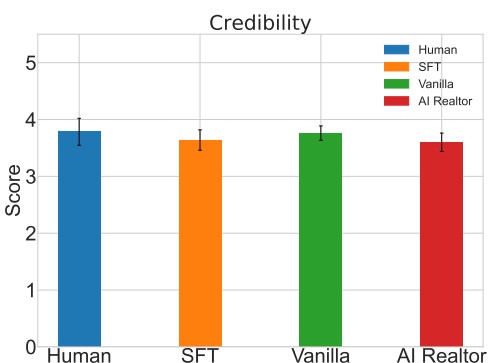

Figure 10: Credibility Scores for Hallucination Checks.

```
Provide a score between 0 and 10, where:
0 = Completely incorrect or irrelevant
5 = Partially correct or relevant
10 = Perfect match

Respond with a JSON object in the following format:
{{
    "score": int
}}

Where 'score' is an integer between 0 and 10.
```

Finally we sum up all scores to compute $\text{Faithful}_{\text{soft}}$.

### E.2 HUMAN EVALUATION

We recruit human annotators to replicate GPT-4o's hallucination checks and assess the reliability of its automatic evaluations. To ensure consistency with the LLM judge, we define factuality identically for human raters: verifying that claims made in the description are strictly grounded in the provided attribute set $X$. In addition to the two factual attributes evaluated by GPT-4o—$X_{\text{hard}}$ and $X_{\text{soft}}$—we include an additional stylistic check: **credibility**, which captures users' emotional judgment of whether the persuasive description feels trustworthy.

Given an attribute set $X$ and a description $L$, either sampled from model- or human-generated outputs, we ask users to (1) rate the credibility of $L$ on a 1–5 scale (Figure 11a), (2) evaluate how well each hard attribute $x_{\text{hard}} \in X_{\text{hard}}$ is reflected in $L$, if it is mentioned ($X_{\text{hard}} \in \text{supp}(L, X_{\text{hard}})$) (Figure 11b), and (3) assess how well each soft attribute $x_{\text{soft}} \in X_{\text{soft}}$ is reflected, if it is mentioned ($x_{\text{soft}} \in \text{supp}(L, X_{\text{soft}})$) (Figure 11c). The instruction files provided to human annotators will be submitted in a separate supplementary file.

As shown in Figure 5, and consistent with findings in § 5.3, 🦝 AI Realtor achieves the highest faithfulness on $X_{\text{hard}}$, while human-written descriptions score lowest in credibility. For evaluations on $X_{\text{soft}}$ (Figure 5) and credibility (Figure 10), which requires more subjective judgment, the performance of 🦝 AI Realtor is comparable to that of humans, suggesting 🦝 AI Realtor does not rely on hallucination or deception to persuade users.

## F PROMPTS

### F.1 KEYWORD EXTRACTION PROMPT

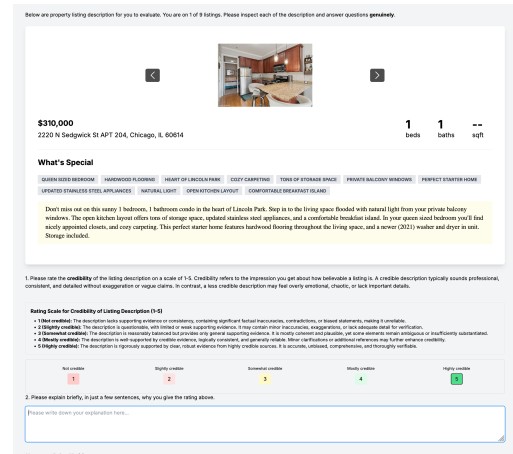

(a) Credibility Evaluation Interface

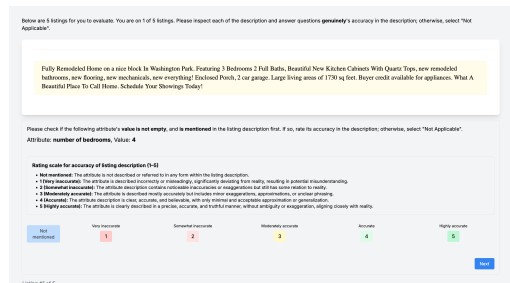

(b) Hard Attribute Evaluation Interface

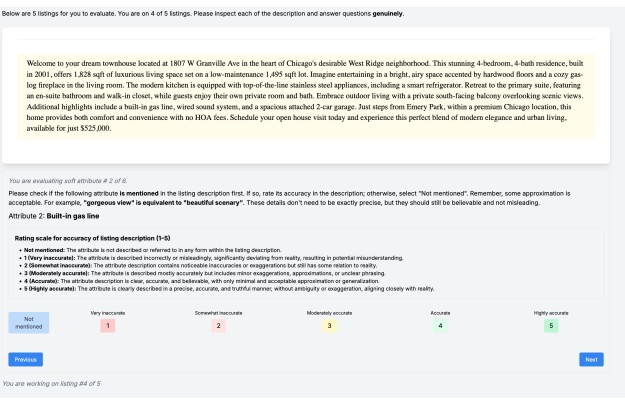

(c) Soft Attribute Evaluation Interface

Figure 11: Interfaces used in the hallucination checks.

```
'Your task is to extract attractive keywords. (e.g., 'modern
 ↪ amenities', 'great views', 'lush landscaping', 'bamboo
 ↪ flooring'). Please express these keywords as phrases or
 ↪ single word from the following house description. Each
 ↪ keyword should be separated by a comma. \n\nDescription: {
 ↪ desc}\n\nKeywords:
```

## F.2 KEYWORD EXTRACTION NORMALIZATION PROMPT

```
"Please remove the quantifiers, numbers, adjectives or any
 ↪ modifiers in the provided input. "
"Uppercase or lowercase doesn't matter. "
"If the given input is already precise enough, please provide
 ↪ the same input."
"If you are not sure what to do, please also provide the input
 ↪  as it is. "
"Do not explain or provide additional information."
"Here are a few examples:"
"\n\nInput: Two Bedrooms.\n\nOutput: Bedrooms."
"\n\nInput: Newly Renovated Kitchen.\n\nOutput: Kitchen."
"\n\nInput: landscape. \n\nOutput: landscape."
"[Example Ends]"
```

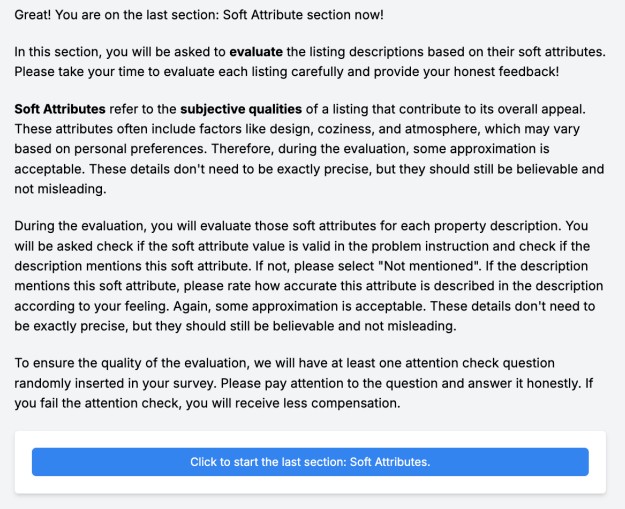

(a) Credibility Evaluation Instruction  (b) Hard Attribute Evaluation Instruction

(c) Soft Attribute Evaluation Instruction

Figure 12: Interfaces used in the hallucination checks.

```
"Now, given the Input, please precisely provide the Output."
"\n\nInput: {}\n\nOutput (should only be a noun phrase or
    ↪ keyword): "
```

### F.3 SCHEMA INDUCTION PROMPT

```
Here is an initial listing keyword schema that I have, but it may
    ↪ not be comprehensive. I have a manually extracted
    ↪ comprehensive keyword list, but there are many duplicated
    ↪ words (e.g., different keywords may bear similar semantic
    ↪ meanings) and some of them may inspire new categories in
    ↪ this schema. I will give you that 1k+ keyword list in a file
    ↪  and the schema below. Can you do it this way: for every 100
    ↪  keywords in the file, either try to assign it to one of the
    ↪  categories below, or create a new (sub)category and assign
    ↪ the keyword to this new (sub)category. You CANNOT use too
    ↪ broad categories like "others" "misc" and "uncategorized".
    ↪ Only create informative categories if necessary. Give me the
    ↪  final zip files containing all 100-ish intermediate
    ↪ assignment results. Each result should be represented as a
    ↪ JSON-like file with key=subcategory, value=[
```

```
        ↪ list_of_original_keywords_in_file], or key=category, value=
        ↪ subcategory (in other words, I want a rich hierarchical
        ↪ structure with the leaf nodes as a list of original keywords
        ↪  in the file).

###schema###
Appliances:
    Refrigerator
    Oven
    Dishwasher
    Washer/Dryer
    Microwave
    Garbage Disposal

Transportation:
    Garage
    Carport
    Parking Space
    Public Transit Nearby

Interior Features:
    Hardwood Floors
    Fireplace
    Central Air Conditioning
    Walk-in Closet
    Open Floor Plan
    High Ceilings

Exterior Features:
    Balcony
    Patio
    Deck
    Fenced Yard
    Garden
    Pool

Building Features:
    Elevator
    Fitness Center
    Laundry Room
    Security System
    Concierge

Utilities:
    Water
    Gas
    Electricity
    Cable/Satellite TV
    Internet

Neighborhood Features:
    Nearby Schools
    Parks
    Shopping Centers
    Restaurants
    Hospitals
    Recreation Facilities
```

## F.4 FEATURE EXTRACTION BASED ON DESCRIPTION PROMPT

```
"Your task is to determine whether the given feature is mentioned
    ↪ in the description. The meaning of the feature will be
    ↪ explained by example keywords. Only respond with 'YES' or '
    ↪ NO'. "
"Feature: {feature_name}. \n\nExample Keywords for explaining this
    ↪  feature: {keywords}\n\n"
"\n\nDescription: {human_description}\n\nResponse (Yes/No): "
```

## F.5 PERSUASIVE LANGUAGE GENERATION WITH PERSONALIZED FEATURES

```
"Your task is to generate a marketing description for a real
    ↪ estate listing with the provided features to highlight, and
    ↪ the client's preferences.
    – The listing has the following attributes:\n{attributes}
    – The listing has the following features (accounted for the
        ↪ client's preference) that are worth highlighting:\n{
        ↪ highlight_features_reweighted }
    – The client has the following general preferences:\n{
        ↪ user_preference}
    – The client has the following specific preferences over
        ↪ features:\n
    {feature_preference}
    – You should emphasize the feature or attributes that matches
        ↪ with the user's preference.
    Make sure the description is persuasive while concise under
        ↪ one paragraph."
```

## F.6 PERSUASIVE LANGUAGE GENERATION WITH LOCALIZED FEATURE PROMPT

```
"Your task is to generate a marketing description for a real
    ↪ estate listing with the provided features to highlight and a
    ↪  list of attributes that are competitive among similar
    ↪ listings."
    – The listing has the following attributes:\n{attributes}
    – Compared with {K} similar listings, the listing stands out
        ↪ in the following features that you want to emphasize:
    {surprisal_features}
    – Compared with listings in Chicago, the following features of
        ↪  this listing are competitive:\n
    {city_rankings}
    – Compared with listings in this neighborhood {neighbourhood},
        ↪  the following features of this listing are competitive
        ↪ :\n
    {neighourhood_rankings}
    – Compared with listings in this zipcode {zipcode}, the
        ↪ following features of this listing are competitive:\n
    {zipcode_rankings}
    – Finally, You should explicitly highlight the listing
        ↪ features or attributes that stands out above or those
        ↪ ones that exactly matches with the user's preferences as
        ↪  a surprise factor.
    Make sure the description is persuasive while concise under
        ↪ one paragraph."
```

## F.7 USER SIMULATION PROMPT

To avoid positional bias as demonstrated in (Zheng et al., 2023), for each pairwise comparisons of descriptions generated by different models, we will prompt the GPT-4o-mini twice to generate separate scores as integers within $[0, 100]$, and compare the final scores to decide which model wins. The prompt below shows an example of this prompt to obtain GPT-4o-mini judgement for the first description presented. "Description 0" and "Description 1" refers to descriptions generated by different models and are randomly shuffled.

```
You will be given a user profile, a listing and two descriptions
    ↪ of this listing. Optionally, you may also be given the user'
    ↪ s history of preferences. Your task is to predict which
    ↪ description the user would prefer. \n\n
User Profile: {user_profile}
Listing: {listing}\n\n
Description 0: {description_0}\n\n
Description 1: {description_1}\n\n
Please first generate an analysis of the user's profile and
    ↪ history (if available), and then analyze why the user might
    ↪ prefer the first description. You can use the following
    ↪ format: 'The user might prefer the first description because
    ↪ ...'
The score for the first description (an integer within [0, 100]):
```

## F.8 RETRIEVER CONFIGURATION

```
"mappings": {
        "properties": {
            "bedrooms": {"type": "float"},
            "bathrooms": {"type": "float"},
            "price": {"type": "float"},
            "description": {"type": "text"},
            "area": {"type": "float"},
            "street_address": {"type": "text"},
            "home_type": {"type": "keyword"},
            "state": {"type": "keyword"},
            "city": {"type": "keyword"},
            "page_view_count": {"type": "float"},
            "favorite_count": {"type": "float"},
            "home_insights": {"type": "keyword"},
            "neighborhood_region": {"type": "keyword"},
            "id": {"type": "keyword"}
        }
    }
```

