# OpenReview forum: "AI Realtor: Towards Grounded Persuasive Language Generation for Automated Copywriting"
_ICLR.cc/2026/Conference — Submitted to ICLR 2026_

### Official Review · Reviewer_amxD · 2025-10-28

**Soundness:** 3
**Presentation:** 3
**Contribution:** 2
**Rating:** 4
**Confidence:** 4

**Summary:**

The paper proposes an agentic framework for grounded, personalized persuasive copy in real estate marketing. The agent has three modules—Grounding (attribute to feature mapping), Personalization (eliciting user preferences), and Marketing (factuality + localized “surprisal” via RAG)—and reports human-subject evaluations where AI Realtor wins ~70% vs. human experts at similar factuality (Elo analysis)

**Strengths:**

* This paper is well-motivated and uses real estate as a concrete, measurable domain for grounded persuasion.
* Personalization design (explicit preference elicitation + selection) and localized “surprisal” via retrieval grounded are novel and plausible mechanisms for persuasion
* The method shows strong gains over human baselines on Real Estate Marketing data

**Weaknesses:**

* The experiment results appear confined to one city/region and a particular property distribution. Persuasion effects are culturally and economically sensitive; cross-market experiment is needed to validate the method's effectiveness. Currently the evidence might be insufficient to show the design generalizes beyond real estate.
* Module ablations show aggregate deltas but not where each module matters. No ablation studies of personalization weight / rarity threshold.
* For a production-adjacent contribution, the paper needs latency, cost, and throughput analysis (for example, per-listing generation time, retrieval budget), and robustness under noisy/missing attributes.

**Questions:**

* How stable are outcomes across cities/price bands and demographics?
* How sensitive are results to $\alpha$ (feature existence), $c$ and $r_{0}$ (personalization strength)?
* Is the SFT method evaluated as a standalone model or a SFT model with the full modular system? Have we tried experiment with smaller model with full modular system?

---

> ### Author Response · Authors · 2025-11-22
> **Responses to Reviewer amxD [1/2]**
>
> We deeply appreciate the reviewer’s thoughtful feedback and their recognition of our work’s motivation and novelty. In particular, your comment on the production-adjacent potential of our approach is insightful, and we agree that our method has meaningful practical implications. At the same time, we would like to gently clarify that this paper is intended as a foundational research study: we introduce the first theory-grounded framework and benchmark for grounded persuasion, supported by an agentic scaffolding explicitly rooted in economic theory. These contributions establish the conceptual and methodological groundwork upon which future production-level systems can be developed. We hope this framing helps clarify the scope and significance of our contribution, and we respectfully invite the reviewer to reconsider the overall assessment in light of its foundational scientific value.
>
> Below we address your comments in details:
>
> **W1 & Q1: On Generalizability and Stability of Outcomes**
>
> > The experiment results appear confined to one city/region... Persuasion effects are culturally and economically sensitive; cross-market experiment is needed to validate the method's effectiveness... How stable are outcomes across cities/price bands and demographics?
>
> This is a very important point. We agree that persuasion is culturally and economically sensitive, and our study is indeed focused on a single market (Chicago) and a specific set of demographics available through the Prolific platform. We chose to focus on a single, yet complex enough market to conduct a deep and methodologically rigorous study, establishing a strong baseline for the novel task of grounded persuasion.
>
> Our agentic workflow is designed to be general, and the core components—such as the attribute-to-feature mapping and the use of RAG for localization—should be readily adaptable to other markets. However, we acknowledge that validating this cross-market and cross-demographic stability is a critical next step. We view our current work as a foundational study that provides a robust methodology and a strong proof-of-concept. We have added a note to the limitations section to explicitly state that further research is needed to validate these findings across different real estate markets and cultural contexts.
>
> **W2 & Q2: On Ablation Studies and Hyperparameter Sensitivity**
>
> > Module ablations show aggregate deltas but not where each module matters. No ablation studies of personalization weight / rarity threshold... How sensitive are results to (feature existence), and (personalization strength)?
>
> Thank you for this suggestion. We agree that a detailed sensitivity analysis of the hyperparameters would be insightful. However, given that our evaluation is based on large-scale human-subject experiments, conducting an exhaustive ablation for each hyperparameter was unfortunately infeasible due to logistical and cost constraints.
>
> The hyperparameters (for feature existence, personalization strength, etc.) were set based on a standard empirical approach: we performed a grid search and evaluated performance on a held-out, human-annotated validation set *before* the main human evaluation study. This process is detailed in Section 4. While this does not provide a full sensitivity analysis, the strong performance of the final system suggests that it is not critically dependent on fine-tuning these specific values. We believe a more detailed analysis of these parameters is an excellent direction for future work, potentially leveraging more scalable (but currently less reliable) AI-based feedback.
>
> **W3: On the Lack of Production-Level Analysis**
>
> > For a production-adjacent contribution, the paper needs latency, cost, and throughput analysis... and robustness under noisy/missing attributes.
>
> We appreciate this perspective. However, we wish to clarify that our primary contribution is a **research framework and benchmark** for studying grounded persuasion, not a production-ready system. Our goal is to establish the feasibility of a theory-grounded agentic approach and to create a reliable methodology for evaluating it. While production-level concerns like latency, cost, and robustness to noisy data are critical for any real-world deployment, they are downstream engineering challenges that fall outside the scope of this foundational research paper. We believe our work provides the scientific basis upon which such production systems can be built.
>
> **That said, a preliminary cost analysis supports this feasibility**: Excluding offline components (e.g., listing retrieval), the production cost is driven by token counts (approx. 3k–4k input, 180–230 output per listing) for full AI Realtor implementation. Considering the efficiency of modern batch inference (e.g., OpenAI API), this workload is negligible compared to standard agentic systems.

---

> ### Author Response · Authors · 2025-11-22
> **Responses to Reviewer amxD [2/2]**
>
> **Q3: On the SFT Method and Smaller Models**
>
> > Is the SFT method evaluated as a standalone model or a SFT model with the full modular system? Have we tried experiment with smaller model with full modular system?
>
> Thank you for this clarifying question. The SFT baseline is a **standalone model**. It was fine-tuned on human-written descriptions to represent a strong, standard approach to this task, without our proposed agentic pipeline. This allows for a direct comparison between our system and a conventional fine-tuning approach.
>
> The experiment using a smaller model (GPT-4o-mini) with our full modular system is already included in our main results (Figure 4). The results show that the agentic pipeline provides a significant boost even to the smaller model, though the larger GPT-4o model still performs best. Using the superior, AI-Realtor-generated descriptions to further fine-tune a smaller model is an excellent idea for model distillation and a promising direction for future work.

---

> > ### Comment · Reviewer_amxD · 2025-11-27
> >
> > I thank the authors for their clarifications and responses. As discussed, the paper builds on an existing economic framework (information design) and applies it to a single, domain-specific setting (real-estate copywriting in Chicago). The contribution is therefore primarily practice-oriented. I would expect thorough end-to-end sensitivity and generalization studies—e.g., across markets, demographics, and data conditions, with comprehensive module and hyperparameter ablations to demonstrate robustness and transferability. These analyses are currently missing, which limits the work’s broader impact beyond this specific application.

---

> ### Author Response · Authors · 2025-12-01
> **Follow-Up Response [1/2]**
>
> We sincerely thank the reviewer for the continued engagement and for clearly articulating the concern regarding the "single, domain-specific setting". While we acknowledge that comprehensive cross-market validation is the ultimate goal of this line of research, we clarify below that our study serves as a **foundational proof-of-concept for the framework of grounded persuasion**.
> 1. **Scope and Feasibility**: This work is designed as a foundational proof-of-concept for a new framework: *grounded persuasion*. Our primary goal is to demonstrate that a theory-driven, agentic approach can achieve superhuman persuasive performance under strict factual constraints. Establishing the feasibility of this task requires rigorous, high-cost human-subject evaluations (participants were compensated $30–$80/hr). These resource constraints limited our ability to perform (1) exhaustive end-to-end hyperparameter sweeps and (2) cross-region validation in this initial study. Despite these resource constraints, our AI Realtor agent notably achieves superhuman performance. This robustly verifies our core hypothesis and demonstrates that the proposed framework is effective, satisfying the primary goal of this research.
>
> 2. **Chicago as a Representative and Diverse Testbed**: We are confident of the generalizability of our method, given that Chicago has a market that offers both national representativeness and high internal diversity. We expanded Section 5.1 in our updated submission PDF to detail this rationale:
>
>     **Representativeness**: Chicago has been established by various economic and sociological literature ([1,2,3]) as a rigorous proxy for broader American urban mechanics. While no single city is perfect, these works suggest Chicago’s demographic and market stratification closely mirror national trends, mitigating concerns that our results are idiosyncratic to a niche market.
>
>     **Diversity (Internal Heterogeneity)**: Chicago has a diverse set of listings, compared to major cities in the US, that can reliably test our models’ performance across various scenarios. As shown in our new analysis (included in the Appendix D.3 of the revision and below as Appendix I), Chicago exhibits a high **Home Type Entropy (0.8613)**, offering a balanced distribution of housing stocks (e.g., 33.7% Single Family, 42.4% Condo, 18.9% Multi-Family) compared to more homogeneous markets like Los Angeles (0.8018) or San Francisco (0.7851).
>
>     **Robust Model Performance**: We also observe that our model successfully tailors its generation across these diverse segments. We have added case studies examples below (also added in Appendix A.4 as case studies in the revision, Appendix II in this thread) illustrating how the model adapts its persuasive strategies for distinct price points (e.g., "distressed/cheap" vs. "luxury/expensive" properties) and housing types.
>
> 3. **Commitment to Open Science for Future Extensions**: We commit to full open-sourcing to enable precisely the broader validation the reviewer calls for. We will publicly release our collected dataset, agent design, and evaluation pipeline. This will allow the community to (1) replicate our methodology in other cultural contexts, (2) conduct deeper hyperparameter analyses, and (3) extend grounded persuasion to domains beyond real estate.
> In summary, while we agree that multi-market generalization is an important future direction, we believe the current paper makes a substantial contribution by formulating the grounded persuasion task and validating it within a market historically proven to be a robust, representative testbed.
>
> **References**
>
> [1] Grabinsky, Jonathan, and Richard V. Reeves. "The Most American City: Chicago, Race, and Inequality." The Brookings Institution, December 21, 2015. https://www.brookings.edu/articles/the-most-american-city-chicago-race-and-inequality/.
>
> [2] Levitt, Steven D., and Chad Syverson. "Market Distortions when Agents are Better Informed: The Value of Information in Real Estate Transactions." The Review of Economics and Statistics 90, no. 4 (2008): 599-611.
>
> [3] Sampson, Robert J. Great American City: Chicago and the Enduring Neighborhood Effect. Chicago: University of Chicago Press, 2012.

---

> ### Author Response · Authors · 2025-12-01
> **Follow-Up Response [2/2]**
>
> ## Appendix I: Home Type Entropy
>
> | City | Home Type Entropy | APARTMENT | CONDO | MANUFACTURED | MULTI_FAMILY | SINGLE_FAMILY | TOWNHOUSE |
> | --- | --- | --- | --- | --- | --- | --- | --- |
> | Chicago, IL | 0.8613 | 0.0% | 42.4% | 0.0% | 18.9% | 33.7% | 4.9% |
> | Seattle, WA | 0.8415 | 0.0% | 24.1% | 0.0% | 4.9% | 49.9% | 21.1% |
> | San Jose, CA | 0.8399 | 0.0% | 25.2% | 7.7% | 8.9% | 47.5% | 10.7% |
> | Los Angeles, CA | 0.8018 | 0.0% | 19.4% | 0.0% | 36.0% | 43.1% | 1.5% |
> | San Francisco, CA | 0.7851 | 0.0% | 45.5% | 0.0% | 17.5% | 35.7% | 1.3% |
>
> The table presents a breakdown of housing types by city, along with the Home Type Entropy for each location. The percentages indicate the proportion of various home types (APARTMENT, CONDO, MANUFACTURED, MULTI_FAMILY, SINGLE_FAMILY, TOWNHOUSE) in each city's real estate market.
>
>
> ## Appendix II: Model Tailoring Examples
>
> We demonstrate some model-generated example descriptions showing that our model can tailor different real estate listings well.
>
> **Core Contrast:** Low-end emphasizes "Safety & Survival" vs. High-end emphasizes "Display & Views"
>
> ###  Low-Price Representative: ($110,000)
> *(Focus: **Defense, Enclosure, Reassurance**. Words aimed at eliminating buyer insecurity regarding the environment.)*
>
>
> $110,000 | 750.0 sqft | 2 beds | 1 bath
>
> Welcome to your charming oasis at [address], nestled in the vibrant and culturally rich Hyde Park neighborhood of Chicago. This inviting 2-bedroom, 1-bathroom condo offers the perfect blend of comfort and convenience at an unbeatable price of $110,000. Step inside to discover a sun-drenched living space adorned with hardwood flooring and an updated kitchen featuring modern appliances, including a wine refrigerator. The thoughtful design includes first-floor conveniences like a full bath and ample storage, with walk-in closets providing plenty of room for your essentials. Enjoy **tranquil moments** on the large back deck, ideal for relaxation or entertaining guests, set within a **gated courtyard that ensures privacy and security**. The property is **meticulously maintained**, boasting brick construction and a welcoming community atmosphere. Although it is **compact**, the space is optimized for comfortable living **without unnecessary upkeep**, perfect for those valuing efficiency. With **proactive security measures**, a **strong sense of community**, and only minutes away from necessities, this condo perfectly encapsulates the ideal home for those **prioritizing safety** and cultural alignment in a vibrant neighborhood.
>
> ---
>
> ### High-Price Representative:  ($1,875,000)
> *(Focus: **Aggression, Openness, Visuals**. Words aimed at showing off transparency and mastery over the environment.)*
>
> $1,875,000 | 0.0 sqft | 4 beds | 4 baths
>
> Discover **unparalleled elegance** and style at [address], a single-family haven nestled in the vibrant Bucktown neighborhood of Chicago. This **exquisite home**, priced at $1,875,000, offers four bedrooms and four bathrooms, perfect for families seeking **ample space and luxury**. Its standout features include a **private corner lot** and a **spacious side yard** designed for **ultimate outdoor enjoyment**, complemented by **gourmet enhancements** like a custom kitchen and a chic beverage center. New Pella windows and **cascading expanses of glass** invite an **abundance of natural light**, creating a bright and airy atmosphere across a versatile loft area ideal for work-from-home needs. With **sophisticated enhancements** such as vaulted ceilings, multiple fireplaces, and a gas fire table, this residence exudes comfort and warmth year-round. The **meticulously crafted design** places this property among the **top tier in architectural style** and elegance within the neighborhood and beyond. Enjoy **seamless access** to essential amenities and natural beauty, with a spacious parking capacity for four cars. **Embrace this rare opportunity** to own a piece of **refined luxury** in an urban yet serene setting.

---

### Official Review · Reviewer_LGBp · 2025-11-01

**Soundness:** 2
**Presentation:** 3
**Contribution:** 2
**Rating:** 4
**Confidence:** 3

**Summary:**

In this work, the authors focus on the task of language generation for grounded persuasion, especially targeting the domain of real estate marketing. For this task, they propose a new benchmark by constructing a corresponding new dataset. Furthermore, to solve the task, they also propose a LLM-based agent on a pipeline that comprises a Grounding Module, mimicking human expertise in identifying and signaling critical, credible selling points, a Personalization Module, tailoring content to user preferences, and a Marketing Module, ensuring factual consistency and incorporating localized features. This pipeline-based workflow is supported by economic theory. Experimental results on the proposed benchmark show the superiority of the proposed pipeline-based agent in win rate and Elo rating against other LLM-based approaches.

**Strengths:**

- This work provides a new benchmark with a new dataset.
- The evaluation is conducted on large-scale human judgment and is thus reliable.
- The authors provide a specific new approach for this task, which is supported by economic theory.
- The proposed approach actually outperforms the baseline in their reliable experiments.
- The ablation study shows the importance of the proposed modules.

**Weaknesses:**

- The target model in the experiment is restricted to GPT-4o and GPT-4o-mini.
- The used baselines are weak. Reasoning models, inference time scaling approaches, and other agent-based approaches are not used.

**Questions:**

- What kind of decoding approach did you use? Greedy (temperature = 0) or sampling (temperature > 0)?
- Do the baseline methods also use RAG in their inference?
- In the SFT baseline, what kind of fine-tuning did you use? (e.g., with or without LoRA)
- In the use of RAG, how many documents are retrieved for each prompt?

Minor comment:
- The equation on p.8 sticks out to the right margin of the paper.

---

> ### Author Response · Authors · 2025-11-22
> **Responses to Reviewer LGBp**
>
> We thank the reviewer for their valuable feedback. We are glad they recognized the reliability of our large-scale human evaluation, the novelty of our proposed benchmark, and the strength of our empirical results.
>
> **W1 & W2: On the scope of models and baselines.**
>
> > The target model in the experiment is restricted to GPT-4o and GPT-4o-mini. The used baselines are weak. Reasoning models, inference time scaling approaches, and other agent-based approaches are not used.
>
> Thank you for these comments. We acknowledge that our experiments are focused on the GPT-4o family of models. This decision was primarily driven by two factors:
>
> 1.  **Methodological Rigor:** Our evaluation relies on a large-scale, controlled human-subject study with over 100 participants. Such studies are logistically complex, time-consuming, and expensive. To ensure the quality and consistency of human feedback, we had to fix the set of models at the start of the study (mid-2024), at which time GPT-4o was the state-of-the-art. Expanding the set of models would have been practically infeasible.
>
> 2.  **Novelty of the Task:** A key contribution of our work is to formulate and establish a rigorous benchmark for a new, challenging task: grounded persuasive language generation. In such a nascent area, there are no established agent-based approaches or specialized reasoning models to directly compare against. Therefore, we selected baselines that represent current, widely-used paradigms: a vanilla LLM (a common approach), a supervised fine-tuned model (SFT), and, most importantly, **expert human professionals**. We respectfully suggest that human experts, who represent the current industry standard, are a very strong and relevant baseline.
>
> Our primary goal was to demonstrate that a principled, theory-grounded agentic workflow could significantly outperform these standard approaches. Now that we have established this benchmark and methodology, we believe our work paves the way for future studies to evaluate a wider range of models and more complex agent architectures.
>
> **Q1: On the decoding approach.**
>
> > What kind of decoding approach did you use? Greedy (temperature = 0) or sampling (temperature > 0)?
>
> We used the default temperature sampling (T=1.0) provided by the OpenAI API for all experiments.
>
> **Q2: On the use of RAG in baselines.**
>
> > Do the baseline methods also use RAG in their inference?
>
> The vanilla and SFT baselines do not use RAG, as they are designed to represent standard, non-agentic approaches. However, we do include an ablation study on our full AI Realtor agent that isolates the effect of the RAG-based Marketing Module. The performance drop in the "w/o Marketing" variant (see Figure 3) demonstrates the value added by the RAG component.
>
> **Q3: On the SFT baseline implementation.**
>
> > In the SFT baseline, what kind of fine-tuning did you use? (e.g., with or without LoRA)
>
> We used the standard fine-tuning API provided by OpenAI. The specifics of their implementation (e.g., use of LoRA) are not publicly disclosed.
>
> **Q4: On the number of retrieved documents in RAG.**
>
> > In the use of RAG, how many documents are retrieved for each prompt?
>
> We retrieve the top 10 most relevant listing descriptions from the surrounding area to identify "surprising" features. We clarify this in Section 4.3 in the revision.
>
> Finally, we will correct the formatting of the equation on page 8. Thank you again for your feedback.

---

### Official Review · Reviewer_5Q3B · 2025-11-01

**Soundness:** 3
**Presentation:** 3
**Contribution:** 2
**Rating:** 6
**Confidence:** 3

**Summary:**

The paper presents AI Realtor, a LLM-based agentic framework that generates persuasive and fact-grounded real-estate contents. The system is decomposed into three modules: grounding module, personalization module and marketing module.

**Strengths:**

1) The topic of this paper is meaningful and practical. The experimental results on the benchmark demonstrate considerable improvements.
2) Overall, the paper demonstrates a high standard in methodology design, experimental scale, and human evaluation, the work is relatively solid.

**Weaknesses:**

1) All three modules are straightforward combinations of existing techniques and the novety is limited.
2) The Hallucination evaluation setting is simple, especially considering the hallucination characteristics of LLM-intrinsic, I do wonder more details of this part.

**Questions:**

See the weakness.

---

> ### Author Response · Authors · 2025-11-21
> **Responses to Reviewer 5Q3B**
>
> We thank the reviewer for their positive feedback on our methodology, experimental scale, and the overall solidity of our work. We are glad the reviewer found the topic meaningful and practical. We address the specific concerns below.
>
> **W1: On the novelty of the proposed method.**
>
> > All three modules are straightforward combinations of existing techniques and the novety is limited.
>
> We appreciate the reviewer’s comment and would like to clarify the novelty of our approach. While our system indeed employs familiar agentic AI tools (e.g., prompting, RAG), we respectfully disagree that it is a straightforward combination of existing techniques. The contribution is not the invention of new primitives, but the principled design, assembly, and grounding of these components through an economic-theoretic lens to address the new task of grounded persuasive content generation.
>
> A central conceptual contribution is our attribute-to-feature mapping mechanism. Inspired by Bayesian persuasion models in information economics, this component formalises how low-level, objective attributes (e.g., a “south-facing window”) are transformed into high-level, value-relevant features (e.g., “abundant natural light”). We propose a principled method to learn this mapping automatically and integrate it tightly into the LLM pipeline. This enables the agent to generate content that is simultaneously factually grounded, persuasively framed, and personalized to consumer preferences—a capability that is not achieved by prior LLM workflows.
>
> Finally, our ablation studies demonstrate that this modular, theory-driven construction provides substantial performance gains, underscoring that the approach is meaningfully more than a simple aggregation of existing techniques.
>
> **W2: On the hallucination evaluation.**
>
> > The Hallucination evaluation setting is simple, especially considering the hallucination characteristics of LLM-intrinsic, I do wonder more details of this part.
>
> Thank you for this question. We recognize the importance of a robust hallucination check, and our evaluation in Section 5.3 is designed to be systematic and fine-grained. Our framework is inspired by previous widely-recognized work FactScore [1] and involves a two-pronged approach:
>
> 1.  **Automated Evaluation:** We use GPT-4o with structured output prompting to conduct a scalable fact-check. We define two types of factual attributes to be checked: **$X_{hard}$** for objective, decision-critical details requiring perfect accuracy (e.g., number of bathrooms, square footage) and **$X_{soft}$** for attributes where minor variations are acceptable (e.g., descriptive home insights). For a given description and each attribute to check, the LLM judge first identifies whether such attribute is mentioned and then scores the faithfulness of this attribute. This provides a more nuanced evaluation than a simple binary check.
>
> 2.  **Human Validation:** To account for the potential biases of LLM-as-a-judge, we validated our automated results with human evaluators. Humans performed the same fact-checking task on a subset of the data. As shown in Figure 5, the human judgments on $X_{hard}$ attributes aligned closely with the automated checker, giving us confidence in our automatic factuality evaluation  approach. Humans also performed an additional "credibility" check to assess whether the descriptions felt trustworthy (Appendix E.2), adding a qualitative dimension to the evaluation.
>
> We believe this multi-faceted approach, combining a scalable automated pipeline with rigorous human validation, provides a detailed and reliable assessment of our model's factuality. The full details and prompts for this process are available in Appendix E.
>
>
> ## References
>
> [1] Min, Sewon, et al. "Factscore: Fine-grained atomic evaluation of factual precision in long form text generation." Proceedings of the 2023 Conference on Empirical Methods in Natural Language Processing. 2023.

---

### Official Review · Reviewer_h9UB · 2025-11-02

**Soundness:** 3
**Presentation:** 4
**Contribution:** 3
**Rating:** 6
**Confidence:** 3

**Summary:**

The paper instantiates a benchmark for persuasive writing inspired by real-estate marketing (REM). REM is chosen because it has (1) rational decision makers, (2) monetary implications for both the buyer and the seller, and (3) structured data sources for grounding. In particular, the authors use real house descriptions from Zillow for the Chicago area and gather annotators who are interested in and rational about buying houses. The task is framed as writing persuasive marketing descriptions for a property that appeal to the annotator's personal preferences.

Two descriptions for the same property written by different sources are presented and pairwise win-rates are collected. Baselines include static human-written descriptions already present on the website, and vanilla LLM and fine-tuned LLM written copy that uses all the attributes of the listing.

AI Realtor on the other hand elicits user preferences with a survey, and identifies listing features that are would appeal to a user. Finally, an LLM (GPT-4o / GPT-4o-mini) is used to write a short description of the property that attempts to persuade the target user by highlighting personalized/stand-out features. AI Realtor includes:
1. Grounding Module to identify abstract property features from the property listing. The paper presents a framework to build this feature space from existing marketing descriptions.
2. Personalization Module to elicit features that are preferred by a particular user and then identifying if such features are present in a given listing.
3. Marketing Module to identify stand-out features about a listing relative to other listings in the region

Results show that user consistently prefer the AI Realtor copy over both the human-written and vanilla LLM-written copy. LLM-as-a-Judge (with human-validation) shows that the Grounding module reduces hallucinations in the AI Realtor written descriptions.

A study is conducted to measure whether LLMs can simulate the users as a means to automate the benchmark. However, initial results suggest that modeling personal preferences is difficult and noisy i.e. in the near term, such benchmarks will need to use real human annotators and cannot be automated.

**Strengths:**

1. The argument for using real-estate marketing (REM) as a testbed for persuasion is very persuasive (caveat in the next section)
    - REM requires grounded persuasion and has measurable economic impact. The source listings are information dense such that persuasion requires selecting the right subset of information to highlight based on the target user
2. The components of AI Realtor are motivated by the real demands of REM
    - The evaluations of factuality show that the AI Realtor can generate persuasive copy grounded in the listed property attributes
    - Marketing and Personalization modules are motivated by criteria known to persuade the audience
3. The potential for simulating the benchmark (and the lack thereof) is valuable information

**Weaknesses:**

1. (Re strength 1) The importance of the description in the real-estate marketing is unclear. Zillow provides exact address, house attributes, and pictures. How important is the descriptive passage in selecting to visit/buy a property compared to the attributes (the pictures for example)?
2. I have certain questions about the definition and evaluation of factuality
    - The extent of information available in the house "attributes" is unclear. So it's hard to understand what inferences are valid and which inferences aren't. Consider the cases below.
    - A statement made by a human realtor about a property (within walking distance to a swimming pool) may be true and demonstrable but may not be explicitly listed in the Zillow house attributes. How is the factuality of such a statement judged? Would the human-written copy be penalized for hallucination?
    - The description in Figure 8 mentions "Minutes from the Lake down Irving". Does this mean that the distance of local landmarks is part of the attributes?
3. ELO is sensitive to the order of comparisons. Please provide confidence intervals on the ELO estimates with bootstrapping (e.g. [LMArena](https://colab.research.google.com/drive/1RAWb22-PFNI-X1gPVzc927SGUdfr6nsR?usp=sharing) provides confidence intervals)

**Questions:**

Clarifications
---
1. Line 429: This line seems to contradict Fig 5 factuality judgements on soft attributes match more than judgements on hard attributes
2. App E2: Human evaluation of factuality measures whether each attribute is faithfully depicted in the description. However, it does not measure whether all features in the description are grounded in the attributes. Thus, there seems to be a mismatch in how factuality is defined for the human raters vs the LLM-as-a-judge

Suggestions
---
Line 83: A single Reddit anecdote cannot be called a "recent report"

---

> ### Author Response · Authors · 2025-11-21
> **Responses to Reviewer h9UB [1/2]**
>
> We thank the reviewer for their positive and constructive feedback. We are glad they found our motivation persuasive and our components well-designed. Below, we address the questions and suggestions.
>
> **W1: On the importance of the property description.**
>
> > The importance of the description in the real-estate marketing is unclear. Zillow provides exact address, house attributes, and pictures. How important is the descriptive passage in selecting to visit/buy a property compared to the attributes (the pictures for example)?
>
> That is an excellent question. While photos and structured attributes are essential for capturing a buyer’s initial attention, the descriptive passage serves a different yet equally important purpose: sustaining interest and conveying the lived experience of the home. According to [Zillow's official guidance for real estate agents](https://www.zillow.com/agents/writing-real-estate-descriptions/), based on their consumer housing reports, a compelling property description is critical for providing insight into what it’s like to live in a home, beyond what photos and attributes alone can provide. Hence, a well-crafted descriptive text could meaningfully contribute to a successful listing. We added this clarification in Section 2 in the revision.
>
>
> **W2: On the definition and evaluation of factuality.**
>
> > I have certain questions about the definition and evaluation of factuality. The extent of information available in the house "attributes" is unclear... A statement made by a human realtor about a property (within walking distance to a swimming pool) may be true and demonstrable but may not be explicitly listed in the Zillow house attributes. How is the factuality of such a statement judged? Would the human-written copy be penalized for hallucination? ... The description in Figure 8 mentions "Minutes from the Lake down Irving". Does this mean that the distance of local landmarks is part of the attributes?
>
> Thank you for these important clarifying questions. For our study, the set of "attributes" for a property consists of all the structured and semi-structured data provided in its Zillow listing. This includes fields like square footage and number of rooms, but also descriptive fields like "home insights" and "neighborhood region," which may contain information about local landmarks. A sample of these attributes is provided in Appendix D.1.
>
> To ensure a rigorous and reproducible evaluation, our factuality check is strictly grounded in this set of attributes. Any statement in a description (whether human- or AI-generated) that is not supported by the information in the Zillow listing is penalized as a potential hallucination. This is a necessary methodological constraint to ensure a fair and controlled comparison. While a human realtor might mention a nearby pool that is not in the listing data, our system cannot verify this external information and thus cannot include it. This standardized approach ensures that both human and AI-generated texts are judged by the same criteria, preventing unverifiable claims and maintaining the integrity of the "grounded persuasion" task.
>
> **W3: On ELO score confidence intervals.**
>
> > ELO is sensitive to the order of comparisons. Please provide confidence intervals on the ELO estimates with bootstrapping (e.g. LMArena provides confidence intervals)
>
> Thank you for this suggestion. We have updated the ELO rating plot in Figure 4 with confidence intervals computed via bootstrapping, as recommended.

---

> ### Author Response · Authors · 2025-11-21
> **Responses to Review h9UB [2/2]**
>
> **Q1: Clarification on factuality judgments (Line 429 vs. Fig 5).**
>
> > Line 429: This line seems to contradict Fig 5 factuality judgements on soft attributes match more than judgements on hard attributes
>
> We appreciate the careful reading. Our apologies for the lack of clarity. We are primarily concerned with the *ranking consistency* between human and LLM judges—that is, whether they agree on the relative factuality of different models (e.g., AI Realtor > Human). Our analysis shows that this relative ranking is more consistent for hard attributes than for soft attributes, even if the absolute agreement scores appear higher for soft attributes. We have revised the text to make this distinction clear.
>
> **Q2: Clarification on factuality definition for human vs. LLM judges (App E2).**
>
> > App E2: Human evaluation of factuality measures whether each attribute is faithfully depicted in the description. However, it does not measure whether all features in the description are grounded in the attributes. Thus, there seems to be a mismatch in how factuality is defined for the human raters vs the LLM-as-a-judge
>
> Thank you for pointing this out. To clarify, our factuality evaluation *only* checks claims against the ground-truth "attributes" from the Zillow listing. The term "features" is part of our agent's internal modeling to interpret and select what to emphasize. However, for the factuality check itself, both human and LLM judges work from the same instructions: verify claims made in the description against the raw attributes. We have revised the appendix to state this more clearly.
>
> Finally, we have also addressed the suggestion regarding the term "recent report" in the manuscript. Thank you again for the helpful and detailed review.

---

### Author Response · Authors · 2025-11-22
**General Responses**

We thank the reviewers for their constructive feedback and encouraging assessment of our work. We are encouraged that the reviewers recognized the **meaningfulness and novelty** of our proposed task (Reviewers h9UB, 5Q3B, amxD, LGBp), the **solidity and scale** of our experiment design and evaluation (Reviewers h9UB, 5Q3B, LGBp), and the **theoretical grounding** of our agentic framework (Reviewers LGBp, h9UB).

We have carefully addressed the reviewers' comments in our individual responses and updated the manuscript (marked in blue). Below, we summarize our response to the common themes raised by the reviewers and outline the key updates to the paper.

### 1. Common Themes & Responses

**On Novelty and Technical Contribution (Reviewers 5Q3B, LGBp)**

Some reviewers questioned whether the method is a straightforward combination of existing tools or suggested broader model comparisons.
*   **Response:** We clarify that our contribution is not merely the aggregation of existing modules, but the **principled design** of an agentic workflow supported by economic theory. Specifically, the *attribute-to-feature mapping*—inspired by Bayesian persuasion models—is a novel mechanism that allows the agent to translate objective data into value-relevant features, a capability not achieved by standard prompting or RAG alone. As Reviewer amxD noted, this establishes a foundational framework for grounded persuasion. We also clarified that our baselines, particularly the **human expert professionals**, represent a rigorous industry standard appropriate for this new benchmark.

**On Evaluation Scope and Generalizability (Reviewers amxD, h9UB)**

Reviewers noted that our study focuses on a single market (Chicago) and requested clarifications on our factuality definitions.
*   **Response:** We acknowledge that this work serves as a **foundational research study**. We focused on a deep, methodologically rigorous human evaluation in a single complex market to establish a reliable benchmark.
*   Regarding factuality, we clarified that our evaluation is strictly constrained to the structured "attributes" provided in the listing (e.g., Zillow data) to ensure reproducibility and fairness between human and AI writers. Claims not supported by these attributes are penalized to prevent unverifiable hallucinations.

### 2. Summary of Paper Updates

We have made the following specific updates to the manuscript (colored in **blue**):
*   **Section 2 Update:** We cite Zillow Annual Report to clarify the importance of descriptions in the real-estate marketing (addressing Reviewer h9UB)
*   **Figure 4 Update:** We added **bootstrapped confidence intervals** to our ELO rating plot to robustly demonstrate the statistical significance of our results (addressing Reviewer h9UB).
*   **Limitations Section:** We explicitly added a note regarding the need for **cross-market and cross-cultural validation**, acknowledging the current focus on the Chicago market (addressing Reviewer amxD).
*   **Section 4.3:** We clarified that the RAG component retrieves the **top 10** most relevant listings (addressing Reviewer LGBp).
*   **Clarifications:** We revised the text to clarify our findings on **ranking consistency** between human and LLM judges (addressing Reviewer h9UB) and fixed the formatting of the equation on page 8 (addressing Reviewer LGBp).
*   **Sections 5.1 & Appendices Update:** We expanded the justification for the Chicago testbed by citing literature on its national representativeness and adding Home Type Entropy analysis (Appendix D.3) and qualitative case studies (Appendix A.4) to demonstrate internal diversity and model robustness (addressing Reviewer amxD).

We believe these updates and clarifications significantly strengthen the paper and thank the reviewers again for their valuable input.

---

### Author Response · Authors · 2025-12-04
**Rebuttal Summary for Area Chair**

We would first like to sincerely thank you for the hard work and extra efforts required to manage the ICLR 2026 review process under these unusual circumstances. We greatly appreciate your time, care, and dedication in ensuring a fair and rigorous evaluation for all submissions. To better assist your review and decision-making process, we provide below a brief summary of our rebuttal and the corresponding manuscript updates for your consideration.

We were encouraged that multiple reviewers independently recognised several key strengths of our work: (1) the significance and novelty of our proposed task (Reviewers h9UB, 5Q3B, amxD, LGBp), (2) the solidity and scale of our experimental design and evaluation (Reviewers h9UB, 5Q3B, LGBp), and (3) the theoretical grounding of our agentic framework (Reviewers LGBp, h9UB).

In the rebuttal, we carefully addressed each reviewer’s concern and incorporated corresponding revisions into the manuscript (marked in blue). We summarise the main points below.

**On novelty and technical contribution**, we clarified that our contribution is not a superficial composition of existing tools, but a principled agentic workflow grounded in economic theory. In particular, we emphasised the attribute-to-feature mapping inspired by Bayesian persuasion as a novel mechanism enabling value-relevant translation from structured data—beyond what standard prompting or RAG alone can achieve. We also clarified the rigor and role of our human-expert baselines as a proper industry-level benchmark.

**On evaluation scope and generalisability**, we clarified that this work is intended as a foundational, methodologically rigorous benchmark, with a deep human-subject evaluation in a single complex market to establish internal validity. We further clarified our strict definition of factuality based solely on structured listing attributes to ensure reproducibility and fair comparison between humans and models, with unsupported claims penalised as hallucinations.

We also made **concrete technical updates**, including:

* Adding bootstrapped confidence intervals to the ELO analysis for statistical robustness;

* Explicitly acknowledging cross-market and cross-cultural limitations;

* Clarifying the RAG retrieval protocol;

* Expanding the justification and internal diversity analysis of the Chicago testbed (entropy analysis and qualitative robustness studies);

While we did not receive many post-rebuttal follow-up comments from reviewers before the discussion phase was cut short, we believe their technical concerns were directly and fully addressed through both our written responses and the corresponding manuscript updates. We therefore hope that our rebuttal and revisions can be fully taken into account when estimating how reviewer impressions might have evolved with additional discussion time.

We sincerely thank the Area Chair again for the careful and thoughtful consideration of our work.

---

### Meta-Review · Area_Chair_M2Fa · 2026-01-07

**Summary:**

This paper reflects a reasonable application of LLMs to an information design problem, focusing on a real-estate setting and a selective disclosure type mechanism.

I agree with reviewer h9UB’s and amxD’s concerns, listed under “Remaining concerns” in the below section. I also have a few concerns the reviewers have not raised. For a few examples:
1. Many mechanisms of copywriting (e.g. framing effects) probably work because receivers (consumers) are not rational Bayesian updaters. The paper does not discuss the fit of the Bayesian persuasion framework and how it relates to these realistic attributes of the copywriting task
2. Even the application of Bayesian persuasion and other Economic ideas seems rather informal. For example, the “commitment” portion of Bayesian persuasion isn’t present, as far as I can tell. The link to signaling theory is not clear to me; what are the *costly* signals here? In general, it’s not clear the theoretical scaffolding adds much of value to the paper
3. For another example, Levitt and Syverson (2008) are cited just for Chicago, but this paper is about agent incentives in real-estate; a very useful thing to discuss that goes basically unmentioned
4. The paper does not evaluate the effects of the proposed system on buyer welfare. Do we know that people are being persuaded for the right reasons, i.e. concrete alignment with their preferences? It’s not clear what efforts are undertaken to prevent manipulation of preferences vs. faithfully implementing them. I.e. we don’t know whether this is helping prospective buyers make *better decisions* which seems like what’s important (e.g. do they overpay, are they satisfied/regretful ex-post, and such). Some evaluation of welfare gains seems important here
5. The paper writes about “…conveying the unique experience of living in a home (Zillow, n.d.). The potential for LLMs to assist in this high-value task…” How might an LLM assist in conveying the unique experience of living in a home? It’s safe to say no LLM has actually done such a thing, and so the motivation for this is lacking without assuming a manipulation-type setting (i.e. tricking the user into believing a fictitious account of life inside that home). I don’t think misinformation is a necessary condition for manipulation, contrary to what is implied by the given ethics statement

Given the informal and incomplete use of theory and the insufficient analysis or discussion of e.g. buyer welfare, I don’t believe the paper offers sufficient scientific insight (vs. commercial value, which is itself weakened by reviewer amxD’s points discussed below).

**Reviewer Concerns:**

### Meaningfully addressed by the author response:
- **h9UB**: W2 factuality (reasonable clarification); W3 CIs (added to the paper); both questions (reasonable clarifications)
- **LGBp**: W1 and W2 are mostly about limitations of the range of models used, which I think the authors have adequately justified; all questions reasonably answered, though some answers are weak without justification (e.g. why temp=1.0? It’s the default is not a good enough reason, this is an important behavioral parameter for GPT-4o)… for Q2, there is nothing “agentic” about RAG by itself
- **amxD**: production analysis (I think this is not necessary for a scientific contribution in principle, and agree with the authors on this front)

### Remaining concerns:
- **h9UB**: W1 importance of description (not rigorously addressed; answer is making a qualitative statement not supported with evidence or analysis sufficient to base the whole paper on);
- **amxD**:  W1/Q1 sensitivity to location (not addressed, but I do acknowledge the challenge), W2 sensitivity analysis (not addressed; the response assumes the only way to do so is in the human participant studies, but in the ML/NLP communities we find proxy metrics routinely to run such studies with to at least get a general idea)

**Reviewer Scores:**

Reviewer amxD seems unlikely to have raised their score given the discussion. Reviewer LGBp and the other two have some cause to increase their score, but it is unclear whether they would.

---

### Decision · Program_Chairs · 2026-01-26

Reject